# DTop-p MoE: Sparsity-Controlled Dynamic Top-p MoE for Foundation Model Pre-training

**Can Jin** [* 1]  **Hongwu Peng** [* 2]  **Mingcan Xiang** [2 3]  **Qixin Zhang** [4]  **Xiangchi Yuan** [5]  **Amit Hasan** [2]  **Ohi Dibua** [2]
**Yifan Gong** [2]  **Yan Kang** [2 †]  **Dimitris N. Metaxas** [1 †]

## Abstract

Sparse Mixture-of-Experts architectures are essential for scaling model capacity efficiently, yet the standard Top-$k$ routing imposes a rigid sparsity pattern that ignores the intrinsic variance in token difficulty and layer-specific computational needs. Top-$p$ routing is more adaptive because it selects experts until their cumulative routing probability reaches a threshold, allowing confident tokens to use fewer experts and ambiguous tokens to recruit more. However, we demonstrate that existing naive Top-$p$ implementations with fixed global probability thresholds provide only marginal gains over Top-$k$, suffer from hyperparameter sensitivity, and result in uncontrolled computational costs. In this paper, we propose DTOP-$p$, a sparsity-controllable dynamic routing mechanism that learns the Top-$p$ probability threshold with a Proportional-Integral controller and uses dynamic routing normalization to support layer-wise expert selection under a global sparsity constraint. Extensive experiments on Large Language Models and Diffusion Transformers demonstrate that DTOP-$p$ consistently outperforms both Top-$k$ and fixed Top-$p$ baselines while matching the average FLOPs of Top-$k$ MoE. Our analysis confirms that DTOP-$p$ exhibits strong scaling properties across expert granularity, total expert capacity, model size, and dataset size, offering a robust and efficient MoE framework for foundation model pre-training.

## 1. Introduction

Scaling model capacity is central to recent progress in Foundation Models (FMs), yet larger dense models incur prohibitive compute and memory costs (OpenAI, 2024; Guo et al., 2025; Liu et al., 2024; OpenAI, 2024; Kaplan et al., 2020; Peebles & Xie, 2023). Sparse Mixture-of-Experts (MoE) architectures mitigate this issue by activating only a small subset of experts per token, thereby decoupling total parameter count from computational cost (Shazeer et al., 2017; Lepikhin et al., 2020; Fedus et al., 2022; Muennighoff et al., 2025; Dai et al., 2024; Wei et al., 2024). As a result, MoE has become a standard design choice for scaling model capacity while keeping training and inference efficient (Yang et al., 2025; Team et al., 2025).

Despite this efficiency, standard Top-$k$ routing enforces a rigid sparsity pattern by selecting a fixed number of $k$ experts regardless of token difficulty or layer-specific needs (Muennighoff et al., 2025; Dai et al., 2024; Fedus et al., 2022; Wei et al., 2024). This design ignores the intrinsic variance of inputs, where complex tokens may require higher capacity (Huang et al., 2024; Jin et al., 2024b), and restricts adaptive resource allocation across depths, as different layers may exhibit varying computational requirements(Jawahar et al., 2019; Dai et al., 2022; Riquelme et al., 2021). Top-$p$ routing is a natural alternative: it sorts experts by routing probability and activates the smallest set whose cumulative probability exceeds a threshold, so high-confidence tokens can use fewer experts while uncertain tokens can access more capacity (Huang et al., 2024; Yang et al., 2024). However, existing Top-$p$ implementations suffer from two critical limitations: (i) they rely on a static global threshold that fails to adapt to evolving training dynamics; and (ii) they yield uncontrolled expert activation, leading to unpredictable computational costs incompatible with strict pre-training compute budgets.

In this work, we demonstrate that fixed-threshold Top-$p$ routing offers only marginal gains over Top-$k$ while suffering from hypersensitivity to $p$ and unstable sparsity. To address these issues, we propose DTOP-$p$, a sparsity-controllable dynamic Top-$p$ routing mechanism. The primary difficulty is that the probability threshold does not naturally receive

---

[*] Equal Contribution, [†] Equal Advising. Work done during an internship at Adobe Research. [1]Rutgers University [2]Adobe Research [3]UMass Amherst [4]Nanyang Technological University [5]Georgia Institute of Technology. Correspondence to: Can Jin <can.jin@rutgers.edu>.

*Proceedings of the 43$^{rd}$ International Conference on Machine Learning*, Seoul, South Korea. PMLR 306, 2026. Copyright 2026 by the author(s).

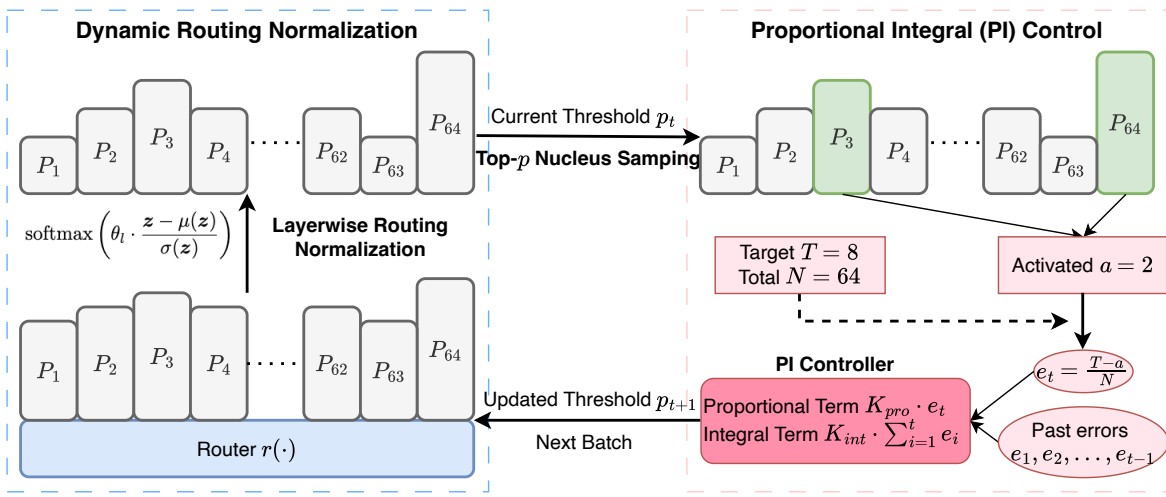

*Figure 1.* Overview of DTOP-$p$ MoE. We employ a Proportional-Integral (PI) controller to dynamically adjust the global probability threshold, aligning the number of activated experts with a target value. The Dynamic Routing Normalization modulates layer-wise logit distributions to support varying sparsity needs, enabling distinct patterns across network depths under the global threshold.

gradients; it merely binarizes the expert mask (0/1), rendering standard optimization ineffective. To overcome this, we apply Proportional–Integral (PI) control from classical control theory (Ziegler & Nichols, 1942; Astrom, 1995). By treating the target sparsity as a setpoint, the controller makes the threshold effectively learnable while meeting the computational budget. To further improve layer-wise routing flexibility, we introduce a dynamic routing normalization that rescales layer-wise routing logits adaptively, enabling flexible expert allocation across layers under the constraint of a learnable global threshold. The overview of DTOP-$p$ MoE is shown in Figure 1.

We conduct extensive experiments on both Large Language Models (LLMs) for Natural Language Processing (NLP) and Diffusion Transformers (DiTs) for Computer Vision (CV). Across benchmarks, DTOP-$p$ consistently outperforms both Top-$k$ and fixed Top-$p$ baselines while matching the average FLOPs of Top-$k$ MoE. We further study scaling behavior with respect to expert granularity, expert capacity, model size, and dataset size, and find that DTOP-$p$ maintains its advantage across a range of scaling regimes. Furthermore, analysis of the activation distribution confirms that DTOP-$p$ adheres to the compute budget while adaptively allocating experts across tokens and layers. Finally, ablations on the PI controller, dynamic routing normalization, architectural choices, and loss designs provide a detailed view of the proposed sparsity-controlled dynamic Top-$p$ mechanism for large-scale pre-training.

Our main contributions are summarized as follows:

★ **Analysis of fixed-threshold Top-$p$ MoE.** We show that fixed Top-$p$ MoE provides limited gains over Top-$k$ while remaining sensitive to the hyperparameter $p$, mak-

ing determining an appropriate $p$ computationally expensive. Furthermore, its uncontrolled sparsity results in unpredictable computational costs, making it unsuitable for large-scale pre-training.

★ **DTOP-$p$ MoE.** We propose a dynamic Top-$p$ MoE that modulates the probability threshold via a PI controller and employs dynamic routing normalization to enable adaptive layer-wise expert selection under a global sparsity constraint. We further study a layer-wise DTOP-$p$ variant for settings requiring explicit per-layer compute control.

★ **Comprehensive empirical investigation.** On NLP and CV benchmarks, DTOP-$p$ achieves better performance and scaling properties compared to baselines. Additional analysis provides robust evidence and practical guidance for adopting DTOP-$p$ in FMs pre-training.

**Conflict of Interest Disclosure.** Several authors are affiliated with Adobe Research, and part of this work was conducted during an internship at Adobe Research. Adobe Research provided research support, computing resources, and internal data access for the CV experiments. To the best of our knowledge, there are no additional financial conflicts of interest to disclose.

## 2. Related Works

Sparse MoE has emerged as a cornerstone architecture for FMs, enabling the scaling of model capacity without a proportional increase in computational cost (Shazeer et al., 2017; Fedus et al., 2022; Yang et al., 2025; Liu et al., 2024). While the standard approach relies on static Top-$k$ routing

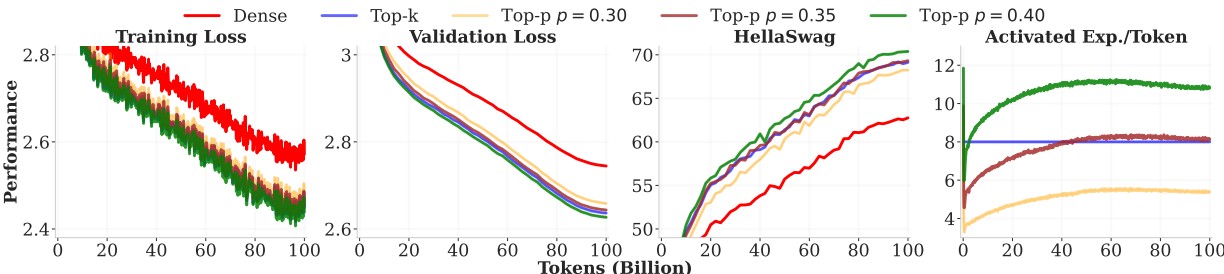

*Figure 2.* Performance comparison of the Dense model, Top-$k$ MoE, and fixed-threshold Top-$p$ MoE ($p \in \{0.30, 0.35, 0.40\}$). Top-$p$ yields only marginal gains over Top-$k$ MoE at comparable activation levels, while the number of activated experts fluctuates unpredictably.

(Lepikhin et al., 2020; Dai et al., 2024), a growing body of research explores dynamic routing mechanisms to better adapt computation to token complexity (Huang et al., 2024; Yang et al., 2024; Guo et al., 2024; Wang et al., 2024). Most relevant to our work are Top-$p$ routing variants (Huang et al., 2024; Yang et al., 2024), which allow for variable expert activation based on cumulative probability. However, these methods typically rely on fixed thresholds, resulting in uncontrolled sparsity. Our approach advances this paradigm by integrating control theory to meet sparsity budgets while maintaining dynamic adaptivity. We provide a more comprehensive review of the literature in Appendix A.

## 3. Preliminary Investigation

To investigate the efficacy of fixed-threshold Top-$p$ MoE, we compare a Dense-1.3B model against an MoE-1.3B-6.9B-64E8A architecture (6.9B total parameters, 1.3B active parameters, 64 total experts, and 8 activated experts; see Table 1 for architecture details). Both models are trained on 100B tokens from the DCLM-Baseline dataset (Li et al., 2024). We evaluate the standard Top-$k$ MoE against the Top-$p$ MoE formulation using fixed thresholds of 0.30, 0.35, and 0.40 (see Section 4 for MoE introduction). All hyperparameters for both Top-$k$ and Top-$p$ MoE are optimized via grid search (see Appendix C for hyperparameter details). Figure 2 illustrates the training/validation/inference performance and the average number of activated experts per token in training batches (batch size ≈ 1M tokens). We observe that: ❶ The number of activated experts in Top-$p$ MoE fluctuates significantly, lacking a consistent trend. Initially, the count drops sharply, then gradually increases, eventually stabilizing near the end of training. This highlights a critical limitation of the current Top-$p$ MoE: *sparsity levels are unpredictable during training*. This unpredictability complicates resource allocation in practical scenarios, as the necessary computational budget is difficult to forecast. Furthermore, the drastic fluctuations in activated experts may induce performance degradation due to gradient instability (Wang et al., 2024). ❷ Regarding performance, Figure 2 shows that only Top-$p$ MoE with a threshold of

0.40 slightly outperforms Top-$k$ MoE; however, it activates over 12 experts for the majority of training. Conversely, with a threshold of 0.35, the model activates over 8 experts yet yields performance merely comparable to Top-$k$ MoE. This suggests that fixed-threshold *Top-p offers only marginal gains over Top-k MoE when activating a similar number of parameters*. ❸ Finally, *Top-p MoE is highly sensitive to threshold selection*. Determining an appropriate threshold requires extensive tuning; an ill-fitted threshold may trigger excessive expert activation in the final training stages, potentially causing Out-of-Memory (OOM) errors and wasting significant computational resources.

## 4. Method

To address the limitations of fixed-threshold Top-$p$ MoE, we propose DTOP-$p$ MoE, which introduces a PI controller to maintain a target sparsity while employing dynamic routing normalization to adaptively adjust logit distributions across layers. We first introduce the concept of Top-$k$ and Top-$p$ MoE and then introduce DTOP-$p$ MoE.

### 4.1. Top-$k$ and Top-$p$ MoE

MoE architectures replace the standard dense feedforward layer in Transformer blocks with an MoE module. This module consists of a set of smaller feedforward networks (FFNs), referred to as experts (Fedus et al., 2022; Lepikhin et al., 2020), and a router—a learned linear layer that maps input representations to specific experts. The output $\boldsymbol{y}$ is the weighted sum of the selected expert outputs:

$$\boldsymbol{y} = \sum_{i=1}^{N} r_i(\boldsymbol{x}) E_i(\boldsymbol{x}) \qquad (1)$$

where $\boldsymbol{x}$ represents the input token representation, $N$ is the total number of experts, $E_i(\cdot)$ denotes the $i$-th expert network, and $r_i(\boldsymbol{x})$ is the routing probability of expert $i$.

Given the router's learnable weight matrix $\boldsymbol{W}$ (omitting bias for simplicity), the initial expert probability distribution

$P(x)$ is computed via the softmax function:

$$P(x) = \text{softmax}(Wx) \qquad (2)$$

In **Top-$k$ MoE** (Fedus et al., 2022; Lepikhin et al., 2020), the routing mechanism selects a fixed number of $k$ experts with the highest probability mass. Let $\sigma$ be a permutation of indices $\{1, \dots, N\}$ that sorts the probabilities in descending order, such that $P_{\sigma(1)}(x) \geq P_{\sigma(2)}(x) \geq \cdots \geq P_{\sigma(N)}(x)$. The routing probabilities for Top-$k$ MoE are:

$$r_i(x) = \begin{cases} \frac{P_i(x)}{\sum_{j=1}^{k} P_{\sigma(j)}(x)} & \text{if } i \in \{\sigma(1), \dots, \sigma(k)\}, \\ 0 & \text{otherwise} \end{cases} \qquad (3)$$

In **Top-$p$ MoE** (Huang et al., 2024; Yang et al., 2024), which adopts the nucleus sampling mechanism (Holtzman et al., 2019), the router selects the smallest set of experts whose cumulative probability exceeds a threshold $p$. We define the dynamic cutoff index $k_p$ as the minimum integer satisfying $\sum_{j=1}^{k_p} P_{\sigma(j)}(x) \geq p$. Then the resulting routing probabilities for Top-$p$ MoE are:

$$r_i(x) = \begin{cases} \frac{P_i(x)}{\sum_{j=1}^{k_p} P_{\sigma(j)}(x)} & \text{if } i \in \{\sigma(1), \dots, \sigma(k_p)\}, \\ 0 & \text{otherwise} \end{cases} \qquad (4)$$

### 4.2. Sparsity-Controlled Dynamic Top-$p$ MoE

#### 4.2.1. PROPORTIONAL-INTEGRAL CONTROL

As shown in Section 3, fixed-threshold Top-$p$ MoE exhibits high computational variance and lacks the explicit resource constraints in Top-$k$ methods. To address these limitations, we propose DTOP-$p$ MoE, which employs a feedback control mechanism to stabilize training. We model the number of activated experts as the process variable and the probability threshold as the control variable, using a PI controller to align the sparsity level with a specified target dynamically.

Formally, given a target number of activated experts $T$, we monitor the average number of activated experts per token, denoted as $a_t$, for the current batch $\mathcal{B}_t$. After computing the routing probabilities via Equation 4, the activation level is:

$$a_t = \frac{1}{M} \sum_{j=1}^{M} \sum_{i=1}^{N} \mathbb{1}(r_i(x_j) > 0) \qquad (5)$$

where $M$ represents the total number of tokens in batch $\mathcal{B}_t$, $N$ is the total number of experts, and $\mathbb{1}(\cdot)$ denotes the indicator function.

Because the threshold $p$ is non-differentiable with respect to the routing decision, it cannot be optimized using gradient descent. Instead, we adjust $p$ between steps to minimize the sparsity error defined as $e_t = (T - a_t)/N$. We apply a

discrete PI control law to update the threshold $p_{t+1}$ for the subsequent batch:

$$p_{t+1} = p_0 + \underbrace{K_{pro} \cdot e_t}_{\text{Proportional}} + \underbrace{K_{int} \cdot \sum_{i=1}^{t} e_i}_{\text{Integral}} \qquad (6)$$

where $p_0$ is the initial probability, and $K_{pro}$ and $K_{int}$ are the proportional and integral gain coefficients, respectively.

This mechanism relies on the monotonicity of Nucleus Sampling (Holtzman et al., 2019): increasing the cumulative probability threshold $p$ strictly requires selecting more experts to satisfy the probability mass requirement. The PI controller uses this correlation to establish a negative feedback loop. A positive error (indicating insufficient activation) increases $p$, compelling the router to select more experts, whereas a negative error reduces $p$. The proportional term addresses immediate deviations, while the integral term accumulates past errors to eliminate steady-state bias (Åström & Murray, 2021). This ensures the model converges to the target $T$ even as the entropy of the logits shifts in training.

#### 4.2.2. DYNAMIC ROUTING NORMALIZATION

Although the PI controller stabilizes global average sparsity, applying a single global threshold $p_t$ across all layers assumes that expert confidence distributions are uniform throughout the network depth. However, deep networks often exhibit distinct sparsity requirements at different layers, such as broader processing in lower layers versus specialized processing in deeper layers (Jawahar et al., 2019; Dai et al., 2022; Riquelme et al., 2021). Applying a global threshold to unnormalized logits may homogenize these patterns, potentially limiting representational capacity.

To address this issue, we introduce *Dynamic Routing Normalization*. Instead of routing directly on raw logits, we normalize the logits and apply a layer-specific learnable scale. This approach decouples the global threshold from the local statistical properties of the logits. For the $l$-th layer, the normalized routing probability is defined as:

$$P(x) = \text{softmax}\left(\theta_l \cdot \frac{z - \mu(z)}{\sigma(z)}\right), \text{ with } z = Wx \qquad (7)$$

where $z$ denotes the raw logits, and $\mu(z)$ and $\sigma(z)$ represent their mean and standard deviation, respectively. The parameter $\theta_l$ is a learnable scalar for layer $l$ that modulates the distribution temperature.

Optimizing $\theta_l$ allows the model to adaptively sharpen or flatten the probability distribution prior to applying the global threshold $p_t$. A larger $\theta_l$ results in a sharper distribution (activating fewer experts), whereas a smaller $\theta_l$ produces a flatter distribution (activating more experts). Consequently,

*Table 1.* Architecture configurations for Dense and MoE variants (including Top-$k$, Top-$p$, and DTOP-$p$ settings) on NLP tasks. Additional configurations and hyperparameters are detailed in Tables 5 and 6 in Appendix C.

| Setting | # Activated/Total Parameters | # Activated/Total Experts | # Layers | # Heads | # Dimension | # FFN Dimension |
|---|---|---|---|---|---|---|
| Dense-0.4B | 0.4B/0.4B | 1/1 | 16 | 16 | 1024 | 4096 |
| Dense-1.3B | 1.3B/1.3B | 1/1 | 16 | 16 | 2048 | 8192 |
| Dense-2.4B | 2.4B/2.4B | 1/1 | 32 | 32 | 2048 | 8192 |
| MoE-0.4B-3.7B-64E8A | 0.4B/3.7B | 8/64 | 16 | 16 | 1024 | 512 |
| MoE-1.3B-2.1B-16E8A | 1.3B/2.1B | 8/16 | 16 | 16 | 2048 | 1024 |
| MoE-1.3B-3.7B-32E8A | 1.3B/3.7B | 8/32 | 16 | 16 | 2048 | 1024 |
| MoE-1.3B-6.9B-32E4A | 1.3B/6.9B | 4/32 | 16 | 16 | 2048 | 2048 |
| MoE-1.3B-6.9B-64E8A | 1.3B/6.9B | 8/64 | 16 | 16 | 2048 | 1024 |
| MoE-1.3B-6.9B-128E16A | 1.3B/6.9B | 16/128 | 16 | 16 | 2048 | 512 |
| MoE-2.4B-13.6B-64E8A | 2.4B/13.6B | 8/64 | 32 | 32 | 2048 | 1024 |

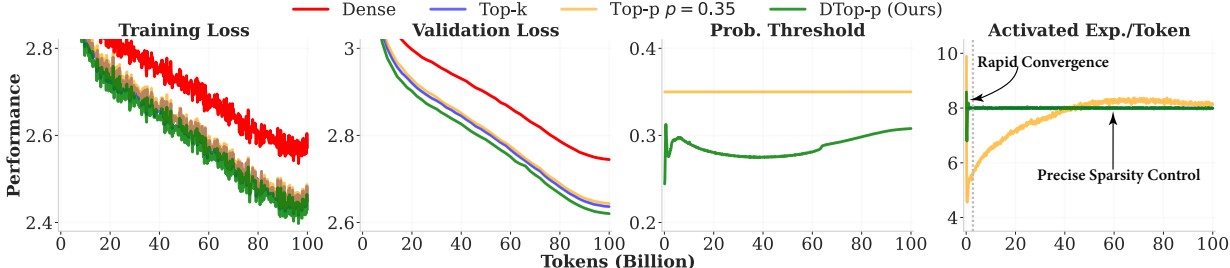

*Figure 3.* Training and validation performance of the Dense-1.3B and MoE-1.3B-6.9B-64E8A models using Top-$k$, Top-$p$, and DTOP-$p$ routing on NLP tasks. DTOP-$p$ achieves the best overall performance.

DTOP-$p$ MoE maintains the global computational budget defined by the PI controller while enabling distinct sparsity patterns for each layer.

### 4.2.3. LAYER-WISE DTOP-$p$ VARIANT

The default DTOP-$p$ uses a single PI-controlled threshold to enforce a global average expert budget while preserving freedom to reallocate computation across layers. For deployment settings that require stricter per-layer predictability, we also consider a layer-wise variant. Instead of one threshold $p_t$, each layer $l$ maintains its own threshold $p_{l,t}$ and PI controller. The average activation of layer $l$ at step $t$ is

$$a_{l,t} = \frac{1}{M}\sum_{j=1}^{M}\sum_{i=1}^{N}\mathbb{1}(r_{l,i}(\boldsymbol{x}_j) > 0), \qquad (8)$$

and the layer-specific threshold is updated by

$$p_{l,t+1} = p_{l,0} + K_{pro}\cdot e_{l,t} + K_{int}\cdot\sum_{i=1}^{t} e_{l,i}, \quad e_{l,t} = \frac{T_l - a_{l,t}}{N}, \qquad (9)$$

where $T_l$ is the target expert budget for layer $l$. In our experiments, we set $T_l = T$ for all layers to match the per-layer compute of Top-$k$ MoE. This variant keeps the token-adaptive nature of Top-$p$ routing but replaces global cross-layer budget sharing with near-deterministic layer-wise budget tracking. The complete training procedure for the default DTOP-$p$ is provided in Algorithm 1 in Appendix

B, with the layer-wise variant obtained by applying the same controller independently to each layer.

## 5. Experiments

In this section, we first outline our experimental setup, including model architectures, datasets, baselines, and hyperparameter configurations. We then evaluate DTOP-$p$ across two domains, NLP and CV, to demonstrate its improved performance over Top-$k$ and fixed-threshold Top-$p$ MoE, particularly regarding performance and precise sparsity control. Beyond establishing main results, we investigate the scaling properties of our method across expert granularity, expert capacity, model size, and training dataset size. Furthermore, we analyze the distribution of activated experts to validate the effectiveness of our sparsity control mechanism and to illustrate the adaptive allocation of experts across layers. Finally, comprehensive ablation studies examine the impact of the PI controller, Dynamic Routing Normalization, and various architectural and loss design choices.

### 5.1. Experimental Details

**Models.** For **NLP** tasks, our primary benchmarks utilize the Dense-1.3B and MoE-1.3B-6.9B-64E8A models. We further investigate the scaling performance using varying expert granularities, capacities, model sizes, and dataset sizes. The model architecture configurations for different settings are detailed in Table 1, with full hyperparameters provided

in Table 5 and Table 6 in the Appendix. To validate DTOP-$p$ in the **CV** domain, we employ a DiT (Peebles & Xie, 2023) featuring the 0.9B dense backbone and the 64E8A MoE variants (0.9B activated and 3.4B total parameters). The architectural details are in Appendix C.

**Datasets.** For **NLP**, models are trained on DCLM-Baseline (DCLM) (Li et al., 2024), a state-of-the-art open-source pre-training corpus filtered from Common Crawl. We train on 100B tokens for the main results and extend to 300B tokens for dataset scaling investigations. Following DCLM, we use the GPT-NeoX-20B (Black et al., 2022) tokenizer with a sequence length of 2048. We monitor validation loss on C4 (Dodge et al., 2021; Raffel et al., 2020) and, following prior works such as OLMoE (Muennighoff et al., 2025) and OLMo (Groeneveld et al., 2024), periodically probe in-training zero-shot performance on HellaSwag (Zellers et al., 2019) and ARC-Easy (Clark et al., 2018). After training evaluation spans **13 datasets** across five skill areas: commonsense reasoning, language understanding, reading comprehension, world knowledge, and symbolic problem solving. For **CV**, models are trained on 2 trillion pixel tokens comprising a mixture of high-quality internal images and image-text pairs. Performance is evaluated via validation loss on internal validation sets. Further details on training and evaluation datasets are in Appendix C and D.

**Baselines.** We compare DTOP-$p$ against three established baselines: (i) *Dense*: A standard dense model serving as a vanilla baseline; (ii) *Top-$k$*: The standard top-$k$ routing MoE; and (iii) *Top-$p$*: A fixed probability threshold Top-$p$ MoE with uncontrolled average expert activation. Note that for Top-$k$ MoE, "activated experts" refers to the fixed count per token, whereas for Top-$p$ and DTOP-$p$ MoE, it denotes the **target** number of activated experts. We focus on token-choice baselines because expert-choice routing requires global token assignment and is less directly applicable to autoregressive LLM pre-training. Methods with auxiliary zero-compute experts are orthogonal to our objective and can be combined with DTOP-$p$; our comparisons isolate whether PI-controlled Top-$p$ routing and DRN can make Top-$p$ compute-controllable and more effective than Top-$k$ under matched FLOPs.

**Hyperparameters.** We select hyperparameters either by following previous works (Muennighoff et al., 2025; Li et al., 2024; Wei et al., 2024) or by identifying optimal settings for each method via grid search. Unless otherwise specified, the same PI coefficients are used across NLP, CV, model-size scaling, and data-size scaling experiments, indicating that the controller does not require model-specific retuning within the evaluated regimes. The detailed values used in our experiments are listed in Table 6 in Appendix C to ensure reproducibility.

*Table 2.* Inference performance comparison between Dense-1.3B and MoE-1.3B-6.9B-64E8A models with Top-$k$, Top-$p$, and DTOP-$p$ MoE trained on 100B tokens. **Bold** indicates the best performance across all settings. Numbers in parentheses indicate the number of few-shot examples used in evaluation. DTOP-$p$ MoE achieves the highest average performance.

| Benchmark | Dense-1.3B | MoE-1.3B-6.9B-64E8A | | |
|---|---|---|---|---|
| | Dense | Top-$k$ | Top-$p$ | DTOP-$p$ (Ours) |
| SVAMP(5) | 5.3 | 10.3 | 8.3 | **16.0** |
| MMLU(5) | 25.2 | 26.6 | 26.8 | **27.4** |
| ARC-Easy(0) | 60.9 | 65.7 | 65.5 | **67.1** |
| ARC-Challenge(0) | 34.4 | 40.6 | 40.9 | **41.7** |
| COPA(5) | 69.0 | 82.0 | 82.0 | **85.0** |
| PIQA(5) | 75.7 | **78.9** | 77.7 | 78.1 |
| HellaSwag(0) | 62.7 | 69.1 | 69.2 | **70.9** |
| WinoGrande(5) | 62.7 | 64.0 | 66.3 | **67.2** |
| LAMBADA(5) | 56.7 | 61.5 | **63.5** | 62.5 |
| BoolQ(5) | 55.4 | 63.9 | 63.9 | **65.4** |
| AGIEval-LSAT-RC(5) | 26.2 | 22.7 | 23.8 | **27.2** |
| AGIEval-LSAT-LR(5) | 26.0 | **26.4** | 24.9 | 25.5 |
| AGIEval-SAT-EN(5) | 26.5 | 24.8 | **27.7** | **27.7** |
| **Average** | 45.1 | 49.0 | 49.3 | **50.9** |

### 5.2. Main Results

**NLP Results.** Figure 3 illustrates the training and validation performance for the Dense-1.3B and MoE-1.3B-6.9B-64E8A models trained on 100B tokens using DTOP-$p$, Top-$k$, and Top-$p$ MoE. The results show that DTOP-$p$ consistently achieves superior training efficiency and lower validation loss. A critical observation is that fixed-threshold Top-$p$ MoE fails to maintain the target sparsity; it frequently activates more than 8 experts during the final training stages, yet yields performance merely comparable to Top-$k$ MoE. In contrast, DTOP-$p$ strictly adheres to the activated expert budget (targeting 8) through the integration of the PI controller and Dynamic Routing Normalization. This ensures that DTOP-$p$ consumes the same average FLOPs as the Top-$k$ baseline while delivering superior performance, demonstrating that DTOP-$p$ effectively balances high performance with computational control.

Table 2 details the inference performance across 13 standard benchmarks. DTOP-$p$ achieves an average improvement of 1.9% over the Top-$k$ MoE baseline. This performance gain indicates that the dynamic allocation of experts significantly enhances the model's reasoning and general knowledge capabilities on complex real-world tasks.

**CV Results.** To verify the generalization of our approach beyond NLP, we conduct experiments using DiTs on CV tasks. Figure 4 compares the training and validation performance of a 0.9B Dense baseline against a 64E8A MoE model (0.9B activated, 3.4B total parameters) trained on 2 trillion pixel tokens. Consistent with our NLP findings, DTOP-$p$ surpasses both the Top-$k$ and Top-$p$ baselines. It

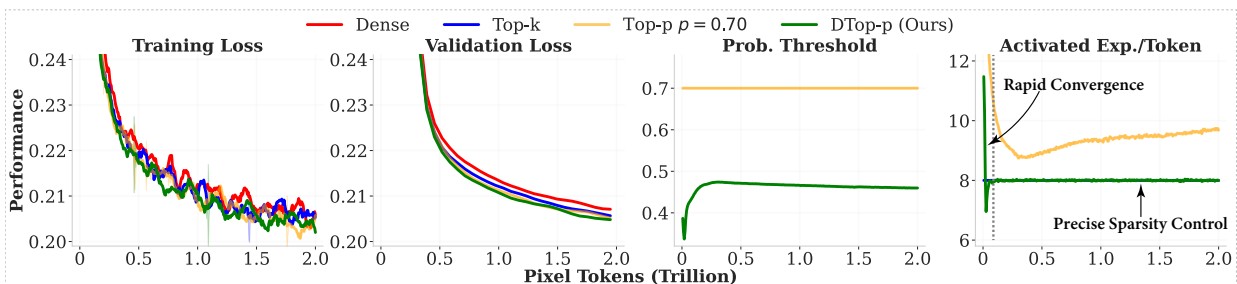

*Figure 4.* Training and validation performance of the 0.9B Dense model versus the 64E8A MoE model (0.9B activated / 3.4B total parameters) using Top-$k$, Top-$p$, and DTOP-$p$ MoE on CV tasks. DTOP-$p$ achieves the best performance.

successfully constrains the number of activated experts to the target level, maintaining equivalent average FLOPs, while achieving superior loss reduction. This confirms DTOP-$p$'s effectiveness and stability in the visual domain.

### 5.3. Scaling Performance

To validate the robustness of DTOP-$p$, we conduct a comprehensive evaluation across four scaling regimes: expert granularity, total expert capacity, model size, and dataset size. In this analysis, we compare our method against Dense and Top-$k$ MoE baselines. We exclude fixed-threshold Top-$p$ MoE, as dynamically tuning probability thresholds to match target sparsity across such a wide range of configurations is computationally intractable. Moreover, our preliminary experiments suggest that fixed-threshold Top-$p$ offers negligible gains over Top-$k$ routing when activated parameters are matched.

**Impact of Expert Granularity.** We first isolate the effect of expert granularity by varying the expert configuration from 4/32 to 16/128 while holding the total and activated parameter counts constant. As detailed in Table 3, DTOP-$p$ demonstrates superior scaling efficiency compared to the baselines. Two key trends emerge: ❶ Performance consistently improves as granularity increases, confirming that our method effectively utilizes finer-grained experts. ❷ The advantage of DTOP-$p$ over Top-$k$ MoE widens at higher granularities (16/128), indicating that our dynamic routing mechanism becomes increasingly beneficial as the routing space grows more complex.

**Impact of Expert Capacity.** Next, we investigate the influence of total expert capacity. In this setting, we expand the total number of experts (increasing total parameters) while fixing the number of activated experts (keeping inference cost constant). The results in Table 4 confirm that DTOP-$p$ successfully leverages the additional capacity. Performance improves monotonically as the total expert count rises, and our method consistently surpasses both Dense and Top-$k$ MoE baselines across all capacity configurations.

*Table 3.* Performance comparison of Dense-1.3B vs. MoE-1.3B-6.9B models across varying expert granularities (32E4A, 64E8A, 128E16A) trained on 100B tokens. DTOP-$p$ scales with expert granularity and outperforms Dense and Top-$k$ MoE baselines.

| Benchmark | Dense-1.3B | 1.3B-6.9B-32E4A | | 1.3B-6.9B-64E8A | | 1.3B-6.9B-128E16A | |
|---|---|---|---|---|---|---|---|
| | Dense | Top-$k$ | DTOP-$p$ | Top-$k$ | DTOP-$p$ | Top-$k$ | DTOP-$p$ |
| SVAMP(5) | 5.3 | 11.6 | 11.0 | 10.3 | **16.0** | 11.6 | 15.0 |
| MMLU(5) | 25.2 | 25.0 | 27.2 | 26.6 | **27.4** | 27.2 | 26.8 |
| ARC-Easy(0) | 60.9 | 67.4 | 66.7 | 65.7 | 67.1 | 66.2 | **69.5** |
| ARC-Challenge(0) | 34.4 | 39.8 | 40.9 | 40.6 | 41.7 | 38.9 | **42.5** |
| COPA(5) | 69.0 | 78.0 | 82.0 | 82.0 | 85.0 | 82.0 | **88.0** |
| PIQA(5) | 75.7 | 77.3 | 78.1 | **78.9** | 78.1 | 77.4 | 78.1 |
| HellaSwag(0) | 62.7 | 68.6 | 69.9 | 69.1 | 70.9 | 69.1 | **71.3** |
| WinoGrande(5) | 62.7 | 64.6 | 65.8 | 64.0 | 67.2 | 64.7 | **67.5** |
| LAMBADA(5) | 56.7 | 62.2 | 63.1 | 61.5 | 62.5 | 62.1 | **64.6** |
| BoolQ(5) | 55.4 | 61.0 | 63.7 | 63.9 | **65.4** | 63.9 | 65.2 |
| AGIEval-LSAT-RC(5) | 26.2 | 24.2 | 26.2 | 22.7 | **27.2** | 24.6 | 26.2 |
| AGIEval-LSAT-LR(5) | 26.0 | 23.3 | 25.3 | **26.4** | 25.5 | 26.0 | 25.9 |
| AGIEval-SAT-EN(5) | 26.5 | 26.8 | 25.9 | 24.8 | 27.7 | 26.8 | **27.9** |
| **Average** | 45.1 | 48.4 | 49.7 | 49.0 | 50.9 | 49.3 | **51.4** |

**Scaling Model and Dataset Size.** Finally, we extend our evaluation to larger backbone architectures and dataset sizes. We vary the model size (0.4B, 1.3B, and 2.4B) and the training dataset size (100B vs. 300B tokens) to assess scalability in realistic pre-training scenarios. The detailed results are provided in Table 8 and Table 9 in Appendix E. Across both dimensions, DTOP-$p$ exhibits strong scaling properties: ❶ It maintains a consistent performance lead over the baselines as model depth and width increase. ❷ It benefits significantly from additional training data, showing that the dynamic routing mechanism remains data-efficient at scale. These results collectively validate DTOP-$p$ as a scalable solution for training FMs.

### 5.4. Activated Experts Distribution Analysis

**Layer-wise Activated Experts Distribution.** We investigate the distribution of activated experts across layers throughout the training process. Figure 5 depicts the activation dynamics for the MoE-1.3B-6.9B-64E8A model trained on 100B tokens, comparing Top-$k$, Top-$p$, and DTOP-$p$ (representative layers are shown; full results are provided in Figure 8 in Appendix F). Two key trends emerge: ❶ In shallow layers ($L_0 - L_2$), the expert activation count in DTOP-$p$ rapidly converges to approximately 1, whereas

*Table 4.* Inference performance of Dense-1.3B vs. MoE models with varying expert capacities (1.3B-2.1B-16E8A, 1.3B-3.7B-32E8A, 1.3B-6.9B-64E8A) trained on 100B tokens. DTop-$p$ achieves stronger performance as total expert capacity increases.

| Benchmark | Dense-1.3B | 1.3B-2.1B-16E8A | | 1.3B-3.7B-32E8A | | 1.3B-6.9B-64E8A | |
|---|---|---|---|---|---|---|---|
| | Dense | Top-$k$ | DTop-$p$ | Top-$k$ | DTop-$p$ | Top-$k$ | DTop-$p$ |
| SVAMP(5) | 5.3 | 6.3 | 8.0 | 5.9 | 11.0 | 10.3 | **16.0** |
| MMLU(5) | 25.2 | 24.8 | 25.8 | 26.7 | 26.7 | 26.6 | **27.4** |
| ARC-Easy(0) | 60.9 | 62.9 | 64.7 | 65.0 | 66.8 | 65.7 | **67.1** |
| ARC-Challenge(0) | 34.4 | 35.4 | 38.1 | 38.9 | 39.2 | 40.6 | **41.7** |
| COPA(5) | 69.0 | 75.0 | 78.0 | 82.0 | **85.0** | 82.0 | **85.0** |
| PIQA(5) | 75.7 | 76.4 | 76.1 | 77.4 | 77.7 | **78.9** | 78.1 |
| HellaSwag(0) | 62.7 | 65.3 | 66.4 | 67.5 | 68.1 | 69.1 | **70.9** |
| WinoGrande(5) | 62.7 | 64.4 | 64.6 | 65.9 | 66.3 | 64.0 | **67.2** |
| LAMBADA(5) | 56.7 | 59.2 | 59.8 | 60.9 | 61.8 | 61.5 | **62.5** |
| BoolQ(5) | 55.4 | 62.0 | 62.7 | 64.9 | 64.1 | 63.9 | **65.4** |
| AGIEval-LSAT-RC(5) | 26.2 | 26.4 | 23.9 | 22.3 | 25.7 | 22.7 | **27.2** |
| AGIEval-LSAT-LR(5) | 26.0 | 23.4 | 23.3 | 24.3 | 25.9 | **26.4** | 25.5 |
| AGIEval-SAT-EN(5) | 26.5 | 23.3 | 23.3 | 23.3 | 23.8 | 24.8 | **27.7** |
| **Average** | 45.1 | 46.5 | 47.3 | 48.1 | 49.4 | 49.0 | **50.9** |

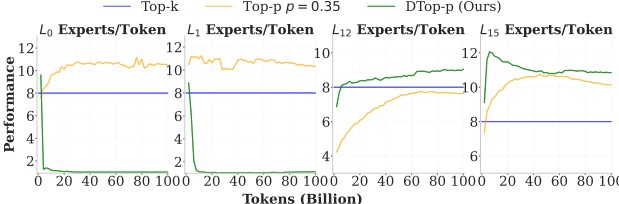

*Figure 5.* Layer-wise evolution of activated experts per token during training (MoE-1.3B-6.9B-64E8A, 100B tokens). We compare Top-$k$, Top-$p$, and DTop-$p$ across 4 representative layers. DTop-$p$ learns to activate fewer experts in shallow layers while utilizing more experts in deeper layers.

fixed-threshold Top-$p$ maintains a consistently high count ($\geq 8$). Conversely, in deeper layers ($L_{12} - L_{15}$), DTop-$p$ utilizes more experts than both Top-$k$ and Top-$p$ baselines. This behavior aligns with prior research suggesting that shallow layers process broad, general features (requiring less capacity), while deeper layers handle specialized semantic reasoning (Dai et al., 2022; Riquelme et al., 2021). This confirms that DTop-$p$ successfully learns distinct, layer-adaptive sparsity patterns. ❷ Additionally, DTop-$p$ achieves convergence faster than Top-$p$ MoE, demonstrating superior stability in activation dynamics.

**Training and Validation Activated Experts.** We further examine global activation statistics to verify the precision of the sparsity control in DTop-$p$. Figure 6 reports the mean and standard deviation of activated experts per token on both the training and validation sets. The results highlight three advantages: ❶ DTop-$p$ precisely maintains the mean number of activated experts at the target value across both datasets, validating the effectiveness of the PI controller. ❷ Activation levels in DTop-$p$ stabilize within the first 1B tokens, whereas Top-$p$ achieves the target sparsity only near the end of training, largely driven by learning rate decay. ❸ DTop-$p$ maintains a moderate and stable standard deviation ($\approx 1$), enabling tokens to flexibly select expert counts based

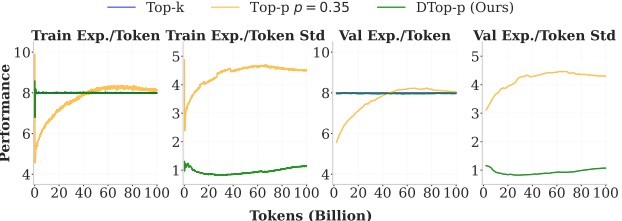

*Figure 6.* Mean and standard deviation of activated experts per token on training and validation sets for Top-$k$, Top-$p$, and DTop-$p$ MoE. DTop-$p$ effectively and rapidly converges to the target activation level on both training and validation datasets.

on difficulty. In contrast, Top-$p$ exhibits high variance ($\approx 4$), indicating unstable routing behavior.

**Layer-wise DTop-$p$.** The layer-wise variant in Section 4.2.3 addresses settings where each layer must obey an explicit compute budget, rather than only the model-wide average. As shown in Figure 9 in Appendix F, layer-wise DTop-$p$ achieves near-perfect tracking of both global and per-layer activated experts while still outperforming Top-$k$ and fixed-threshold Top-$p$ under matched FLOPs. Its validation performance is slightly below the default global-threshold DTop-$p$, suggesting a clear trade-off: per-layer controllers improve deployment predictability and bandwidth planning, whereas the default design preserves more cross-layer freedom and can allocate additional experts to deeper, more specialized layers when useful.

### 5.5. Ablation Studies

**Ablation of DTop-$p$ Components.** We evaluate the individual contributions of the PI controller and Dynamic Routing Normalization (DRN) through ablation experiments on the MoE-1.3B-6.9B-64E8A model trained on 30B tokens. We compare DTop-$p$ against three baseline configurations: (1) *Top-$p$ w/o both*: A standard fixed-threshold Top-$p$ MoE ($p = 0.25$) utilizing fixed global routing logit normalization with a temperature of 1, following Wei et al. (2024); (2) *Top-$p$ w/o PI*: Incorporates DRN but relies on a fixed threshold ($p = 0.25$); and (3) *Top-$p$ w/o DRN*: Utilizes the PI controller combined with fixed global routing logit normalization (temperature 1). The results, detailed in Figure 7, reveal the following: ❶ DTop-$p$ achieves optimal performance only when the PI controller and DRN are combined. ❷ The PI controller is critical for strictly enforcing the expert budget; without it, the number of activated experts remains unregulated and tends to increase uncontrollably during the early stages of training. ❸ DRN further enhances performance by adaptively rescaling layer-wise logit distributions. This mechanism stabilizes the probability threshold and routing behavior, effectively mitigating the severe threshold fluctuations observed when DRN is excluded.

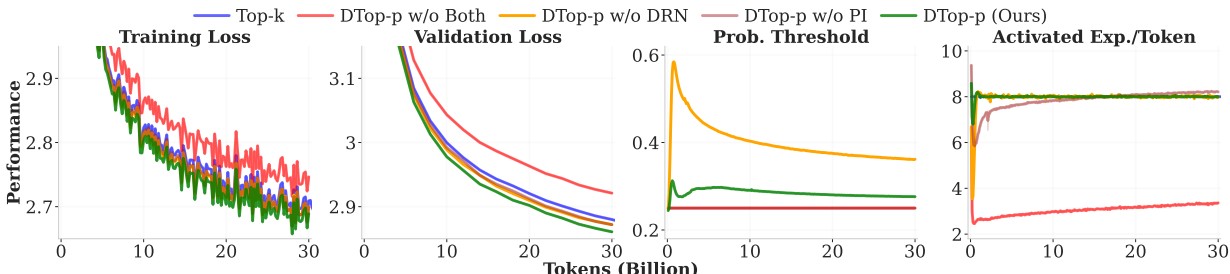

*Figure 7.* Ablation study of the PI controller (PI) and Dynamic Routing Normalization (DRN).

Additionally, we compare DTOP-$p$ MoE against a Top-$k$ MoE baseline augmented with DRN (see Figure 10 in Appendix F). The analysis shows that: ❶ DTOP-$p$ MoE consistently outperforms Top-$k$ MoE with DRN. ❷ While DRN improves Top-$k$ MoE, it yields a significantly larger gain for Top-$p$ MoE. This indicates that DRN is particularly effective when the number of activated experts varies adaptively, rather than being constrained to a uniform $k$.

**PI Controller Tuning.** We examine the sensitivity of the PI controller to its hyperparameters, specifically the probability initialization, $K_{pro}$, and $K_{int}$, using the MoE-1.3B-6.9B-64E8A model trained on 20B tokens. The results are presented in Figure 11 (probability initialization) and Figure 12 (PI coefficients) in Appendix F. We observe two key trends: ❶ DTOP-$p$ demonstrates high robustness to probability initialization values. The PI controller rapidly aligns the number of activated experts with the target level regardless of the initial setting. This is a significant advantage over fixed-threshold Top-$p$ MoE, which is highly sensitive to initialization and requires extensive tuning to identify an appropriate threshold (as illustrated in Figure 2). ❷ The coefficients $K_{pro}$ and $K_{int}$ govern the trade-off between convergence speed and stability. Moderate values yield optimal performance, ensuring rapid convergence of activated experts while maintaining a stable probability threshold. Excessively large values induce threshold instability, which degrades performance, whereas very small values result in slow convergence.

## 6. Additional Investigations

To further optimize DTOP-$p$, we conduct extensive ablations regarding architectural choices, optimizer hyperparameters, and loss function designs. Due to space constraints, we provide a summary of these findings below and refer readers to Appendix G for a comprehensive analysis. **Weight Tying** (Appendix G.1). We evaluate the impact of sharing parameters between the input token embedding and the output projection layer. As shown in Figure 13, we find that weight tying significantly improves the performance of DTOP-$p$ compared to the untied configuration, likely due to

the implicit regularization it provides. **AdamW Optimizer Tuning** (Appendix G.2). We perform a comprehensive hyperparameter sweep for the AdamW optimizer, analyzing Learning Rate (LR), Weight Decay, and Epsilon ($\epsilon$). Results in Figures 14, 15, and 16 indicate that a moderate LR (3e-3) and Weight Decay (3.3e-2) yield optimal convergence. Notably, smaller $\epsilon$ values (1e-8) consistently outperform larger ones, suggesting that a lower $\epsilon$ enhances numerical stability and adaptivity when handling the dynamic routing distributions inherent to DTOP-$p$. **Loss Function Design** (Appendix G.3). We optimize a composite objective function comprising Language Modeling Loss (Eq. 10), Load Balancing Loss (Eq. 11), Dynamic Loss (Eq. 13), and Router Z-Loss (Eq. 14). Our ablation studies in Figures 17, 18, 19, and 20 reveal that while Load Balancing and Dynamic losses are essential for preventing router collapse and encouraging decisiveness, Router Z-Loss proves detrimental. By penalizing large logits, the Z-Loss forces the routing distribution to become excessively flat, hindering the experts' ability to specialize. Consequently, we exclude Router Z-Loss from our final training objective.

## 7. Conclusion

In this work, we propose DTOP-$p$ to address the rigidity of Top-$k$ selection and the instability of fixed-threshold Top-$p$ MoE. By synergizing a PI controller with dynamic routing normalization, DTOP-$p$ enables adaptive expert allocation across tokens and layers while meeting the global computational budget. Extensive experiments on NLP and CV benchmarks demonstrate that our method consistently outperforms baselines and exhibits robust scaling properties across expert granularity, total expert capacity, model size, and dataset size. These results establish DTOP-$p$ as a robust, scalable solution that successfully reconciles dynamic adaptivity with strict computational constraints for foundation model pre-training.

## Impact Statement

Our work focuses on improving the computational efficiency of sparse Mixture-of-Experts models, aiming to contribute

to the broader goal of sustainable Green AI. We acknowledge, however, that architectural efficiency does not mitigate the inherent biases or toxicity found in web-scale training data like DCLM-Baseline. As with any general performance advancement, we emphasize that this method should be coupled with rigorous safety alignment and red-teaming to address potential dual-use risks prior to deployment.

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

## A. Detailed Related Works

**Mixture-of-Experts.** Mixture-of-Experts (MoE) traces back to work in the early 1990s as a way to decompose a task into multiple specialized experts coordinated by a router. Shazeer et al. (2017) introduce a sparse MoE layer between LSTM blocks and report strong results on language modeling and machine translation. Fedus et al. (2022); Lepikhin et al. (2020) integrate sparse MoE into Transformers and adopt a Top-$k$ routing strategy for expert selection. As model scale grows, MoE becomes a prevailing approach for increasing capacity in foundation models without increasing activated computation (Yang et al., 2025; Guo et al., 2025; Bai et al., 2025). Several works focus on making expert selection more fine-grained or more stable (Dai et al., 2024; Lepikhin et al., 2020), on analyzing and improving scaling behavior of MoE models (Ludziejewski et al., 2024), and on improving training efficiency through better systems and kernels (Lepikhin et al., 2020; Gale et al., 2023). Another active line studies alternative routing mechanisms beyond fixed Top-$k$ (Wei et al., 2024; Jin et al., 2024b; Huang et al., 2024; Yang et al., 2024; Wang et al., 2024; Guo et al., 2024; Zhou et al., 2022). Zhou et al. (2022) study expert choice instead of token choice. Jin et al. (2024b) explore adding zero-compute experts to MoE. Wang et al. (2024); Guo et al. (2024) develop dynamic or differentiable routing to adapt expert assignment to the input. Huang et al. (2024); Yang et al. (2024) investigate accumulative Top-$p$ routing with a fixed probability threshold. Our work stays closest to this dynamic/Top-$p$ group, but introduces a *sparsity-controlled* dynamic Top-$p$ MoE: we make the probability threshold effectively learnable via a PI controller and pair it with a dynamic routing normalization mechanism so that layers can adaptively learn different routing patterns under a single global sparsity budget.

**Foundation Models.** Large foundation models (LFMs) show strong generality across NLP, CV, and multimodal tasks (OpenAI, 2024; Guo et al., 2025; Liu et al., 2024; OpenAI, 2024; Peebles & Xie, 2023; Bai et al., 2025; Jin et al., 2024a; 2025; Zhou et al., 2025b; Liu & Zhu, 2022; 2023; 2024; Zhou et al., 2025a), mainly because model size and data size continue to scale. Dense LLMs with billions of parameters already reach competitive performance on many benchmarks (Brown et al., 2020; Ouyang et al., 2022; Grattafiori et al., 2024; Yang et al., 2025; Yao et al., 2025; Cai et al., 2025; Li et al., 2025; Qu et al., 2026; Zou et al., 2025; Wang et al., 2025). At the same time, MoE-based LFMs become the dominant design in frontier systems (Yang et al., 2025; Team et al., 2025; Liu et al., 2024; Guo et al., 2025; OpenAI, 2025), since they match or surpass dense models at similar activated compute and keep improving when model size, data size, expert granularity, and total expert capacity increase (Ludziejewski et al., 2024; Muennighoff et al., 2025).

## B. Algorithm

Algorithm 1 outlines the training procedure for DTOP-$p$. During the forward pass, the router first applies Dynamic Routing Normalization (Equation 7) to adaptively scale the logits within each layer. Subsequently, experts are selected via nucleus sampling based on the current global probability threshold $p_t$. To maintain the computational budget, the PI controller acts as a feedback mechanism: it monitors the deviation between the actual average number of activated experts $a_t$ and the target $T$. This sparsity error is used to update the threshold $p_{t+1}$ for the next step, leveraging proportional and integral terms to ensure the sparsity level converges stably to the target. For layer-wise DTOP-$p$, the same update is applied independently for each layer using the layer-specific statistics in Equations 8 and 9. Finally, all model parameters are optimized via gradient descent.

## C. Training Details

**NLP Tasks.** The main results utilize the Dense-1.3B and MoE-1.3B-6.9B-64E8A models trained on 100B DCLM-Baseline tokens. Architecture configurations are detailed in Tables 1 and 5, with method-specific hyperparameters provided in Table 6. In all experiments, we employ high expert granularity (e.g., 64E8A) rather than limited sets (e.g., 4E1A or 8E1A) to maximize the efficacy of the MoE architecture (Dai et al., 2024; Ludziejewski et al., 2024). Scaling performance investigations use the main hyperparameters unless otherwise specified. For ablation studies, we vary only the specific parameter under investigation relative to the DTOP-$p$ main hyperparameters listed in Table 6. We use the DCLM-Baseline dataset (Li et al., 2024), a filtered subset of Common Crawl containing 3.8T tokens, from which we randomly sample 20B to 300B tokens for our experiments. We utilize the GPT-NeoX-20B (Black et al., 2022) tokenizer with a sequence length of 2048. Due to resource constraints, most ablation experiments are conducted on 30B tokens. Validation is performed on the C4 validation set. All experiments are conducted using 64 or 128 NVIDIA A100 80GB GPUs.

---

**Algorithm 1** DTOP-$p$ MoE

---

**Require:** Dataset $\mathcal{D}$, target expert $T$, initial probability threshold $p_0$, PI gain coefficient $K_{pro}$, $K_{int}$, model parameters $\Theta$ (including dynamic scales $\{\theta_l\}_{l=1}^L$)

1: Initialize integral error accumulation $e_{sum} \leftarrow 0$
2: Initialize probability threshold $p_1 = p_0$
3: **for** step $t = 1, 2, \ldots$ with batch $\mathcal{B}_t \in \mathcal{D}$ **do**
4:     Accumulator for number of activated experts $a_{sum} \leftarrow 0$
5:     **Forward Pass:**
6:     **for** layer $l = 1$ to $L$ **do**
7:         Compute raw logits for input representation $\boldsymbol{x}$: $\boldsymbol{z} \leftarrow \boldsymbol{Wx}$
8:         **Dynamic Routing Normalization** (Equation 7): $\boldsymbol{P} \leftarrow \text{softmax}\left(\theta_l \cdot \frac{\boldsymbol{z} - \mu(\boldsymbol{z})}{\sigma(\boldsymbol{z})}\right)$
9:         **Nucleus Sampling** with global threshold $p_t$:
10:           Select minimal set of experts $S$ such that $\sum_{i \in S} P_i \geq p_t$
11:         $r_i(\boldsymbol{x}) \leftarrow \frac{P_i}{\sum_{j \in S} P_j}$ for $i \in S$, else 0 (Equation 4)
12:         Record number of activated experts: $a_{sum} \leftarrow a_{sum} + |S|$
13:         Compute layer output: $\boldsymbol{y} \leftarrow \sum_{i \in S} r_i(\boldsymbol{x}) E_i(\boldsymbol{x})$
14:     **end for**
15:     **PI controller Update** (Equation 6):
16:     Calculate average activation per token: $a_t \leftarrow a_{sum}/(L \cdot |\mathcal{B}_t|)$, $|\mathcal{B}_t|$ is total tokens in $\mathcal{B}_t$
17:     Calculate sparsity error: $e_t \leftarrow (T - a_t)/N$
18:     Update integral term: $e_{sum} \leftarrow e_{sum} + e_t$
19:     Update global threshold:
20:         $p_{t+1} \leftarrow p_0 + K_{pro} \cdot e_t + K_{int} \cdot e_{sum}$
21:     Clip $p_{t+1}$ to range $(0, 1)$
22:     **Optimization:**
23:     Compute Total Loss $\mathcal{L}$
24:     Update parameters $\Theta$ via gradient descent
25: **end for**

---

**CV Tasks.** We employ a DiT-style denoiser comprising 28 layers, a hidden size of 2048, and 16 attention heads. Training utilizes the AdamW optimizer with a learning rate of $1 \times 10^{-4}$ and weight decay of 0.01. We apply a Load Balancing (LB) Loss coefficient of $1 \times 10^{-6}$ and a Dynamic Loss coefficient of $1 \times 10^{-5}$, omitting the Router Z-loss as it proves less critical for DiT pre-training. We set the PI controller initial probability to 0.40; the Dynamic Routing Normalization and other PI controller hyperparameters match those used in the NLP experiments. Models are trained on 2 trillion pixel tokens aggregated from high-quality internal image datasets and image-text pairs. Performance is evaluated via validation loss on the internal validation set. All experiments are conducted using 64 NVIDIA A100 80GB GPUs.

## D. NLP Tasks Evaluation Details

**Setup.** During pre-training, we monitor validation loss on C4 (Dodge et al., 2021; Raffel et al., 2020) and, following prior work such as DataComp-LM (Li et al., 2024), OLMoE (Muennighoff et al., 2025), and OLMo (Groeneveld et al., 2024), periodically probe **zero-shot** in-training performance on HellaSwag (Zellers et al., 2019) and ARC-Easy (Clark et al., 2018). After pre-training, we conduct a comprehensive evaluation on **13 datasets** spanning five skill areas: commonsense reasoning, language understanding, reading comprehension, world knowledge, and symbolic problem solving. Unless otherwise specified, multiple-choice tasks are reported with **accuracy** (Acc.); the random baseline is 25% for four-choice and 50% for two-choice. For free-response tasks, we report **exact match** (EM). Final evaluations use a **0/5-shot** setting with the standard community splits (see Table 7 for more details).

**World Knowledge.** This field probes encyclopedic and academic knowledge across the sciences, mathematics, social sciences, and humanities; items resemble standardized exams with carefully designed distractors.

- **MMLU** (Hendrycks et al., 2021) (14,042 items; 4-choice): 57 subjects (e.g., jurisprudence, mathematics, ethics).

*Table 5.* Architecture configurations for the Dense and MoE models on NLP tasks.

| Architecture | Dense | MoE |
|---|---|---|
| Activation | SwiGLU | SwiGLU |
| Vocab size | 50,432 | 50,432 |
| LayerNorm type | Low-precision LayerNorm (Team, 2021) | Low-precision LayerNorm |
| LayerNorm $\epsilon$ | $1 \times 10^{-5}$ | $1 \times 10^{-5}$ |
| QK-Norm | Yes | Yes |
| Position embedding | RoPE | RoPE |
| Biases | – | – |
| Weight tying | Yes | Yes |
| Init distribution | Truncated normal | Truncated normal |
| Init std | 0.02 | 0.02 |
| Init truncation | $3\times$ std | $3\times$ std |
| MoE layers | Every | Every |

*Table 6.* Training hyperparameters for Dense, Top-$k$ MoE, Top-$p$ MoE, and DTOP-$p$ MoE on NLP tasks ("–" indicates not used/applicable).

| Hyperparameters | Dense | Top-$k$ MoE | Top-$p$ MoE | DTOP-$p$ MoE |
|---|---|---|---|---|
| Sequence Length | 2,048 | 2,048 | 2,048 | 2,048 |
| Batch Size (Samples) | 512 | 512 | 512 | 512 |
| Batch Size (Tokens) | $\sim$1M | $\sim$1M | $\sim$1M | $\sim$1M |
| Warmup Steps | 2,500 | 2,500 | 2,500 | 2,500 |
| Peak LR | 3.0E-3 | 3.0E-3 | 3.0E-3 | 3.0E-3 |
| Minimum LR | 3.0E-5 | 3.0E-5 | 3.0E-5 | 3.0E-5 |
| Optimizer | AdamW | AdamW | AdamW | AdamW |
| Weight Decay | 3.3E-2 | 3.3E-2 | 3.3E-2 | 3.3E-2 |
| Beta1 | 0.9 | 0.9 | 0.9 | 0.9 |
| Beta2 | 0.95 | 0.95 | 0.95 | 0.95 |
| AdamW Epsilon | 1.0E-8 | 1.0E-8 | 1.0E-8 | 1.0E-8 |
| LR Schedule | Cosine | Cosine | Cosine | Cosine |
| Gradient Clipping | Global 1.0 | Global 1.0 | Global 1.0 | Global 1.0 |
| Gradient Reduce Dtype | FP32 | FP32 | FP32 | FP32 |
| Optimizer State Dtype | FP32 | FP32 | FP32 | FP32 |
| LM Z-loss Coefficient $\lambda_z$ | 1.0E-4 | 1.0E-4 | 1.0E-4 | 1.0E-4 |
| LB Loss Coefficient $\lambda_{lbl}$ | – | 1.0E-4 | 1.0E-4 | 1.0E-4 |
| Dynamic Loss Coefficient $\lambda_{dl}$ | – | – | 1.0E-3 | 1.0E-3 |
| Router Z-loss Coefficient $\lambda_{rz}$ | – | 0 | 0 | 0 |
| Probability Threshold Init. | – | – | 0.25/0.30/0.35/0.40 | 0.25 |
| Routing Normalization $\theta_l$ | – | Global 1.0 (Wei et al., 2024) | Global 1.0 | Layer-wise Dynamic |
| PI $K_{pro}$ Coefficient | – | – | – | 0.1 |
| PI $K_{int}$ Coefficient | – | – | – | 0.1 |

Given a question and four options, the model outputs A/B/C/D. Metric: Acc.; random baseline **25%**.

- **ARC-Easy** (Clark et al., 2018) (2,376 items; 4-choice): Grade 3–9 science questions emphasizing basic scientific knowledge. Metric: Acc.; random baseline **25%**.

- **ARC-Challenge** (Clark et al., 2018) (1,172 items; 4-choice): The harder ARC subset requiring non-trivial science knowledge and procedural reasoning. Metric: Acc.; random baseline **25%**.

**Symbolic Problem Solving.**   Benchmarks discrete, symbolic reasoning such as arithmetic and algebraic manipulation.

- **SVAMP** (Patel et al., 2021) (300 problems; free-response): Short grade-school arithmetic word problems with numerical answers. We prompt for a chain-of-thought before the final answer and score with EM. Random baseline **0%**.

**Commonsense Reasoning.**   Evaluates everyday causal and physical knowledge needed to select plausible outcomes.

- **COPA** (Gordon et al., 2012) (100 items; 2-choice): Given a premise, choose the more plausible cause or effect. Metric: Acc.; random baseline **50%**.

- **PIQA** (Bisk et al., 2020) (1,838 items; 2-choice): Physical commonsense—select the more sensible solution to a basic task. Metric: Acc.; random baseline **50%**.

**Language Understanding.**    Assesses discourse coherence, referential consistency, and general linguistic regularities.

- **HellaSwag** (Zellers et al., 2019) (10,042 items; 4-choice): Given a partial narrative, choose the most likely continuation. Metric: Acc.; random baseline **25%**.

- **WinoGrande** (Sakaguchi et al., 2021) (1,267 items; 2-choice): Large-scale coreference challenge—fill a blank with one of two candidates so the sentence is semantically valid. Metric: Acc.; random baseline **50%**.

- **LAMBADA** (Paperno et al., 2016) (2016; 5,153 passages; LM/EM): Read a passage and predict its final token; scored with EM. Random baseline **0%**.

**Reading Comprehension.**    Tests whether a model can answer questions using information explicitly or implicitly present in a passage.

- **BoolQ** (Clark et al., 2019) (3,270 items; yes/no): Short passages followed by binary questions. Metric: Acc.; random baseline **50%**.

- **AGIEval-LSAT-RC** (Zhong et al., 2024) (268 items; 4-choice): LSAT Reading Comprehension—extract and integrate factual information from passages. Metric: Acc.; random baseline **25%**.

- **AGIEval-LSAT-LR** (Zhong et al., 2024) (510 items; 4-choice): LSAT Logical Reasoning—draw conclusions and identify assumptions from short arguments. Metric: Acc.; random baseline **25%**.

- **AGIEval-SAT-EN** (Zhong et al., 2024) (206 items; 4-choice): SAT-style English questions targeting high-school level reading and grammar. Metric: Acc.; random baseline **25%**.

*Table 7.* Evaluation suites and settings used during and after pre-training for NLP tasks.

| Stage | Field | Dataset | Split | #Samples | Shot | Metric | Random Acc. |
|---|---|---|---|---|---|---|---|
| *During Pre-training* | Language Understanding | HellaSwag (Zellers et al., 2019) | val | 10,042 | 0 | Multiple Choice Acc. | 25% |
| | World Knowledge | ARC-Easy (Clark et al., 2018) | test | 2,376 | 0 | Multiple Choice Acc. | 25% |
| *After Pre-training* | World Knowledge | MMLU (Hendrycks et al., 2021) | test | 14,042 | 5 | Multiple Choice Acc. | 25% |
| | World Knowledge | ARC-Easy (Clark et al., 2018) | test | 2,376 | 0 | Multiple Choice Acc. | 25% |
| | World Knowledge | ARC-Challenge (Clark et al., 2018) | test | 1,172 | 0 | Multiple Choice Acc. | 25% |
| | Symbolic Problem Solving | SVAMP (Patel et al., 2021) | test | 300 | 5 | Exact Match Acc. | 0% |
| | Commonsense Reasoning | COPA (Gordon et al., 2012) | val | 100 | 5 | Multiple Choice Acc. | 50% |
| | Commonsense Reasoning | PIQA (Bisk et al., 2020) | val | 1,838 | 5 | Multiple Choice Acc. | 50% |
| | Language Understanding | HellaSwag (Zellers et al., 2019) | val | 10,042 | 0 | Multiple Choice Acc. | 25% |
| | Language Understanding | WinoGrande (Sakaguchi et al., 2021) | val | 1,267 | 5 | Cloze Formulation Acc. | 50% |
| | Language Understanding | LAMBADA (Paperno et al., 2016) | test | 5,153 | 5 | Language Modeling Acc. | 0% |
| | Reading Comprehension | BoolQ (Clark et al., 2019) | val | 3,270 | 5 | Multiple Choice Acc. | 50% |
| | Reading Comprehension | AGIEval-LSAT-RC (Zhong et al., 2024) | test | 268 | 5 | Multiple Choice Acc. | 25% |
| | Reading Comprehension | AGIEval-LSAT-LR (Zhong et al., 2024) | test | 510 | 5 | Multiple Choice Acc. | 25% |
| | Reading Comprehension | AGIEval-SAT-EN (Zhong et al., 2024) | test | 206 | 5 | Multiple Choice Acc. | 25% |

# E. Scaling Performance Details

In this section, we provide detailed experimental results regarding the scalability of DTop-$p$ across two critical dimensions: model size and training dataset size. Due to space constraints in the main text, the comprehensive performance breakdowns for these regimes are presented here.

**Scaling Model Size.**    We evaluate the scalability of DTop-$p$ by varying the model architecture, specifically the number of layers, attention heads, and hidden dimensions, while maintaining a fixed expert granularity. The comparative results for 0.4B, 1.3B, and 2.4B parameter models are detailed in Table 8. Two key observations emerge from this analysis:

- **Consistent Superiority:** DTOP-$p$ consistently achieves the highest average performance across all three model sizes, outperforming both Dense and Top-$k$ MoE baselines.

- **Improved Scaling Efficiency:** The performance advantage of DTOP-$p$ over Top-$k$ MoE becomes more distinct as the model size increases. For instance, on the 2.4B model, DTOP-$p$ achieves a significant lead in reasoning benchmarks such as SVAMP and COPA, suggesting that the dynamic routing mechanism effectively leverages the increased capacity of larger architectures.

*Table 8.* Inference performance of Dense models vs. varying 64E8A MoE models across different model sizes (0.4B, 1.3B, 2.4B) trained on 100B tokens. DTOP-$p$ consistently achieves the best performance and scales effectively with model size.

| Area | Benchmark | Dense-0.4B | MoE-0.4B-3.7B | | Dense-1.3B | MoE-1.3B-6.9B | | Dense-2.4B | MoE-2.4B-13.6B | |
|---|---|---|---|---|---|---|---|---|---|---|
| | | Dense | Top-$k$ | DTOP-$p$ | Dense | Top-$k$ | DTOP-$p$ | Dense | Top-$k$ | DTOP-$p$ |
| *Symbolic Problem Solving* | SVAMP(5) | 6.3 | 4.0 | 6.6 | 5.3 | 10.3 | 16.0 | 11.0 | 13.6 | **22.6** |
| *World Knowledge* | MMLU(5) | 25.0 | 23.9 | 24.8 | 25.2 | 26.6 | 27.4 | 24.6 | 28.2 | **29.0** |
| | ARC-Easy(0) | 52.9 | 61.3 | 62.1 | 60.9 | 65.7 | 67.1 | 64.7 | 70.2 | **70.7** |
| | ARC-Challenge(0) | 26.9 | 32.5 | 33.7 | 34.4 | 40.6 | 41.7 | 37.7 | **45.5** | 44.6 |
| *Commonsense Reasoning* | COPA(5) | 63.0 | 72.0 | 74.0 | 69.0 | 82.0 | 85.0 | 80.0 | 84.0 | **87.0** |
| | PIQA(5) | 71.2 | 74.5 | 75.0 | 75.7 | 78.9 | 78.1 | 77.2 | 79.1 | **79.5** |
| *Language Understanding* | HellaSwag(0) | 48.8 | 60.6 | 60.7 | 62.7 | 69.1 | 70.9 | 67.9 | 73.9 | **74.4** |
| | WinoGrande(5) | 55.8 | 59.7 | 59.9 | 62.7 | 64.0 | 67.2 | 67.4 | 70.6 | **71.8** |
| | LAMBADA(5) | 46.8 | 54.5 | 56.2 | 56.7 | 61.5 | 62.5 | 63.8 | 68.6 | **69.4** |
| *Reading Comprehension* | BoolQ(5) | 58.9 | 62.4 | 64.1 | 55.4 | 63.9 | 65.4 | 69.2 | 68.0 | **71.2** |
| | AGIEval-LSAT-RC(5) | 21.2 | 23.9 | 23.9 | 26.2 | 22.7 | **27.2** | 24.2 | 24.2 | 26.2 |
| | AGIEval-LSAT-LR(5) | 24.5 | 22.1 | 23.5 | 26.0 | **26.4** | 25.5 | 24.5 | 22.1 | 26.1 |
| | AGIEval-SAT-EN(5) | 23.7 | 22.3 | 23.3 | 26.5 | 24.8 | 27.7 | 27.6 | 27.7 | **30.5** |
| **Average** | | 40.4 | 44.1 | 45.2 | 45.1 | 49.0 | 50.9 | 49.2 | 52.0 | **54.1** |

**Scaling Dataset Size.** We further investigate the impact of training data volume by scaling the dataset from 100B to 300B tokens, utilizing the same model architecture with 1.3B activated parameters. Table 9 details the performance comparison. The results confirm the data efficiency of our approach:

- **Positive Data Scaling:** DTOP-$p$ exhibits strong scaling properties with respect to data size. As the training tokens increase, the model demonstrates significant improvements across world knowledge and reading comprehension benchmarks.

- **Sustained Advantage:** At the 300B token scale, DTOP-$p$ maintains its superiority over both Dense and Top-$k$ MoE counterparts. This validates that the sparsity-controlled dynamic routing mechanism remains robust and effective even as the model is exposed to larger and more diverse data distributions.

## F. Detailed Analyses and Ablation Studies

**Layer-wise Activated Experts Distribution.** The full results for layer-wise expert activation across all 16 layers are presented in Figure 8. These comprehensive plots corroborate the findings discussed in Section 5.4: DTOP-$p$ consistently adopts a hierarchical sparsity strategy. Specifically, the model assigns minimal computational resources (nearing a single expert) to early layers ($L_0 - L_2$) to handle generic inputs, while allocating a larger expert budget to deeper layers ($L_{12} - L_{15}$) to manage complex semantic specializations.

**Layer-wise DTOP-$p$ Budget Control.** Figure 9 compares Top-$k$, fixed-threshold Top-$p$, default DTOP-$p$, and layer-wise DTOP-$p$. The layer-wise variant controls each layer with an independent PI threshold and therefore removes the large per-layer budget variation observed in the default global-threshold design. It nearly matches the target activated experts in every layer while maintaining the same global average FLOPs as Top-$k$. The result also clarifies the trade-off discussed in Section 5.4: strict per-layer control improves deployment predictability and bandwidth planning, but slightly underperforms the default DTOP-$p$ because it reduces the model's ability to shift computation from shallow to deeper layers.

*Table 9.* Inference performance of Dense-1.3B vs. MoE-1.3B-6.9B-64E8A models trained on 100B tokens versus 300B tokens. DTOP-$p$ consistently achieves the best performance and scales effectively with training dataset size.

| Area | Benchmark | Dense-1.3B (100B) | MoE-1.3B-6.9B (100B) | | Dense-1.3B (300B) | MoE-1.3B-6.9B (300B) | |
|---|---|---|---|---|---|---|---|
| | | Dense | Top-$k$ | DTOP-$p$ | Dense | Top-$k$ | DTOP-$p$ |
| *Symbolic Problem Solving* | SVAMP(5) | 5.3 | 10.3 | 16.0 | 6.3 | 22.6 | **27.7** |
| *World Knowledge* | MMLU(5) | 25.2 | 26.6 | 27.4 | 25.3 | 27.7 | **29.1** |
| | ARC-Easy(0) | 60.9 | 65.7 | 67.1 | 64.2 | 70.5 | **71.2** |
| | ARC-Challenge(0) | 34.4 | 40.6 | 41.7 | 37.4 | 44.1 | **45.3** |
| *Commonsense Reasoning* | COPA(5) | 69.0 | 82.0 | 85.0 | 79.0 | 82.0 | **86.0** |
| | PIQA(5) | 75.7 | 78.9 | 78.1 | 77.1 | 79.1 | **79.8** |
| *Language Understanding* | HellaSwag(0) | 62.7 | 69.1 | 70.9 | 66.3 | 73.9 | **74.5** |
| | WinoGrande(5) | 62.7 | 64.0 | 67.2 | 64.9 | 69.2 | **70.4** |
| | LAMBADA(5) | 56.7 | 61.5 | 62.5 | 61.8 | 64.5 | **68.2** |
| *Reading Comprehension* | BoolQ(5) | 55.4 | 63.9 | 65.4 | 66.4 | 70.5 | **71.9** |
| | AGIEval-LSAT-RC(5) | 26.2 | 22.7 | **27.2** | 25.7 | 23.8 | 25.7 |
| | AGIEval-LSAT-LR(5) | 26.0 | **26.4** | 25.5 | 23.9 | 22.1 | 24.9 |
| | AGIEval-SAT-EN(5) | 26.5 | 24.8 | 27.7 | **28.6** | 25.2 | 25.2 |
| **Average** | | 45.1 | 49.0 | 50.9 | 48.2 | 51.9 | **53.8** |

**Ablation of DTOP-$p$ Components.** Figure 10 presents the ablation results for Top-$k$, Top-$k$ with Dynamic Routing Normalization (DRN), DTOP-$p$, and DTOP-$p$ variants (without the PI controller, without DRN, and without both). We observe that DTOP-$p$ achieves optimal performance when both the PI controller and DRN are utilized. Furthermore, DTOP-$p$ outperforms Top-$k$ paired with DRN, demonstrating the effectiveness of the combined PI controller and DRN mechanisms.

*Table 10.* Final training and validation losses of Dense, Top-$k$, Top-$p$, and DTOP-$p$ MoE on NLP and CV from Figures 3 and 4.

| Loss | Dense | Top-$k$ | Top-$p$ | DTOP-$p$ (Ours) |
|---|---|---|---|---|
| NLP-Train | 2.5714 | 2.4495 | 2.4520 | **2.4285** |
| NLP-Validation | 2.7445 | 2.6384 | 2.6445 | **2.6193** |
| CV-Train | 0.2058 | 0.2053 | 0.2051 | **0.2035** |
| CV-Validation | 0.2073 | 0.2058 | 0.2055 | **0.2048** |

**PI Controller Tuning.** This paragraph presents the supplementary visualizations for the PI controller hyperparameter tuning experiments discussed in Section 5.5. Figure 11 illustrates the training trajectories across varying probability initialization values, confirming the robustness of DTOP-$p$ MoE. Figure 12 depicts the impact of different proportional ($K_{pro}$) and integral ($K_{int}$) coefficients on the convergence speed and stability of the expert activation in DTOP-$p$ MoE.

# G. Detailed Additional Investigations

## G.1. Weight Tying Analysis

Weight tying unifies the parameter space by sharing weights between the input token embedding layer and the final output projection layer, whereas untied approaches maintain distinct parameters for these components. To evaluate the efficacy of this technique, we compare DTOP-$p$ with and without weight tying using the MoE-1.3B-6.9B-64E8A architecture trained on 30B tokens.

As illustrated in Figure 13, the configuration utilizing weight tying achieves significantly lower validation loss than its untied counterpart. We hypothesize that this improvement stems from two factors: (1) the implicit regularization introduced by reducing the total number of trainable parameters, which helps mitigate overfitting; and (2) the alignment of the input and output embedding spaces, which facilitates more stable convergence by enforcing geometric consistency.

## G.2. AdamW Optimizer Tuning

We conduct a comprehensive hyperparameter sweep for the AdamW optimizer, specifically analyzing the Learning Rate (LR), Weight Decay, and Epsilon. The performance for the MoE-1.3B-6.9B-64E8A model trained on 100B tokens are detailed in Figures 14, 15, and 16). Our findings indicate: ❶ **Learning Rate:** A moderate LR of 3e-3 yields optimal performance. Excessively high LRs induce instability, evidenced by significant spikes in training loss, whereas smaller LRs lead to premature plateaus at higher loss levels, indicating a failure to fully traverse the loss landscape within the training budget. ❷ **Weight Decay:** A moderate Weight Decay of 3.3e-2 proves most effective. Higher values appear to over-regularize the model, impeding the reduction of training loss. Conversely, insufficient weight decay (smaller values) results in suboptimal convergence. ❸ **Epsilon:** Smaller Epsilon values (1e-8 and 1e-10) consistently outperform larger ones for DTOP-$p$. This aligns with observations in Top-$k$ MoE models (Muennighoff et al., 2025), suggesting that lower Epsilon values can maintain numerical stability and improve the optimizer's adaptive properties when handling the dynamic routing distributions.

## G.3. Loss Designs

To ensure training stability and routing efficiency in DTOP-$p$, we optimize a composite objective function for NLP tasks comprising four distinct terms: Language Modeling Loss, Load Balancing Loss, Dynamic Loss, and Router Z-Loss. To determine the optimal configuration, we conduct comprehensive hyperparameter tuning using the MoE-1.3B-6.9B-64E8A model trained on 30B tokens. Results for these ablation studies are shown in Figures 17, 18, 19, and 20.

**Language Modeling Loss (LM Loss).**  Our primary optimization objective is the standard Cross-Entropy Loss, augmented with an auxiliary Z-Loss term (Chowdhery et al., 2023) to enhance training stability. While the Cross-Entropy term minimizes the negative log-likelihood of the target tokens, the Z-Loss regularizes the partition function of the final softmax layer. This regularization mitigates "logit drift"—a phenomenon where the magnitude of output logits increases indefinitely, leading to numerical instability. The total language modeling loss is defined as:

$$L_{\text{LM}} = \frac{1}{M} \sum_{j=1}^{M} \left[ \underbrace{-\log \frac{\exp(\boldsymbol{z}_{j,y_j})}{\sum_{v=1}^{V} \exp(\boldsymbol{z}_{j,v})}}_{\text{Cross-Entropy}} + \lambda_z \cdot \underbrace{\left( \log \sum_{v=1}^{V} \exp(\boldsymbol{z}_{j,v}) \right)^2}_{\text{Auxiliary Z-Loss}} \right] \tag{10}$$

where $M$ is the batch size (in tokens), $V$ is the vocabulary size, $\boldsymbol{z}_{j,v}$ denotes the output logit for the $j$-th token and vocabulary index $v$, $y_j$ is the ground-truth token index, and $\lambda_z$ is the LM Z-Loss coefficient.

**Observation:** As shown in Figure 17, the LM Z-Loss has a minor impact on DTOP-$p$ performance. A small coefficient of $1 \times 10^{-4}$ yields the best results.

**Load Balancing Loss (LB Loss).**  Widely adopted in sparse MoE architectures (Shazeer et al., 2017; Lepikhin et al., 2020), the Load Balancing Loss prevents the router from collapsing into a trivial solution where only a few experts are utilized. It encourages a uniform distribution of tokens across all experts. The loss is formulated as:

$$L_{\text{LB}} = \lambda_{lbl} \cdot N \cdot \sum_{i=1}^{N} f_i \cdot Q_i \tag{11}$$

where $\lambda_{lbl}$ is the LB Loss coefficient and $N$ is the number of experts. $f_i$ denotes the fraction of tokens discretely dispatched to expert $i$, and $Q_i$ represents the average probability mass allocated to expert $i$:

$$f_i = \frac{1}{M} \sum_{j=1}^{M} \mathbb{1}(r_i(\boldsymbol{x}_j) > 0), \quad Q_i = \frac{1}{M} \sum_{j=1}^{M} P_i(\boldsymbol{x}_j) \tag{12}$$

Here, $\boldsymbol{x}_j$ is the input representation for the $j$-th token, and $\mathbb{1}(\cdot)$ is the indicator function.

**Observation:** As shown in Figure 18, a small coefficient of $1 \times 10^{-4}$ achieves optimal performance. Larger coefficients degrade performance, likely because excessive weight on auxiliary balancing constraints interferes with the primary optimization of model weights.

**Dynamic Loss.** To encourage decisiveness in the router, we employ the Dynamic Loss (Huang et al., 2024). This objective minimizes the entropy of the routing distribution, thereby compelling the router to assign high probabilities to a small, decisive set of experts (sharpening the distribution) rather than spreading probability mass uniformly. It is defined as:

$$L_{\text{Dynamic}} = -\lambda_{dl} \cdot \frac{1}{M} \sum_{j=1}^{M} \sum_{i=1}^{N} P_i(\boldsymbol{x}_j) \cdot \log(P_i(\boldsymbol{x}_j)) \tag{13}$$

where $\lambda_{dl}$ is the Dynamic Loss coefficient.

**Observation:** As shown in Figure 19, a moderate coefficient of $1 \times 10^{-3}$ achieves the best performance. Larger coefficients negatively impact results by making the routing distribution excessively sharp, which increases the probability threshold too aggressively and hinders exploration.

**Router Z-Loss.** Finally, to ensure numerical stability within the MoE layers, we adopt the Router Z-Loss introduced in ST-MoE (Zoph et al., 2022). Large input logits can cause numeric overflow during the large-scale matrix multiplications characteristic of MoE layers. The Router Z-Loss penalizes large logit magnitudes:

$$L_{\text{Router-Z}} = \lambda_{rz} \cdot \frac{1}{M} \sum_{j=1}^{M} \left( \log \sum_{i=1}^{N} \exp(\boldsymbol{W}_i \cdot \boldsymbol{x}_j^T) \right)^2 \tag{14}$$

where $\lambda_{rz}$ is the Router Z-Loss coefficient and $\boldsymbol{W}_i \cdot \boldsymbol{x}_j^T$ represents the pre-softmax logit for the $i$-th expert.

**Observation:** As shown in Figure 20, Router Z-Loss proves counter-productive for DTOP-$p$. Increasing the coefficient causes the probability threshold to drop dramatically, indicating that Router Z-Loss forces the routing distribution to become flatter. This flattening prevents experts from learning distinct, specialized knowledge (Huang et al., 2024; Wei et al., 2024). Consequently, we set $\lambda_{rz} = 0$.

## H. Practical Deployment Discussions

**Distributed Synchronization Overhead.** DTOP-$p$ introduces only a lightweight scalar feedback signal beyond standard token-choice MoE training. The controller needs the average number of activated experts per token, which can be computed from the same token-expert assignment tensor used by the load-balancing loss. Thus, the required all-reduce can be fused with existing MoE statistics collection, and the threshold update is performed between optimization steps.

**Load Balancing and Dynamic Routing.** Dynamic routing and load balancing operate at different levels. The PI controller constrains how many experts are activated on average, while DRN controls how the routing distribution changes across tokens and layers. The load-balancing loss regularizes expert-level utilization inside each MoE layer, preventing expert collapse even when different tokens or layers use different numbers of experts. Therefore, these mechanisms are complementary: DTOP-$p$ enables adaptive token- and layer-level compute allocation while retaining regularized expert-level utilization.

**Controller Stability Under Distribution Shifts.** The integral term reduces steady-state bias by accumulating past sparsity errors, but large integral gains can cause threshold oscillation. We mitigate this in three ways: the sparsity error is normalized by the total number of experts, the threshold is clipped to $(0, 1)$, and we use moderate PI gains that are stable across NLP, CV, model-size scaling, and data-size scaling experiments. With large pre-training batches (around one million tokens), abrupt full-batch distribution shifts are also smoothed statistically. If a deployment requires even stricter layer-level compute predictability, layer-wise DTOP-$p$ provides independent per-layer controllers as discussed above. Automated gain adaptation is a promising future extension, but our current experiments suggest that fixed moderate gains are already robust across the evaluated settings.

**Baseline Scope.** Our main comparisons focus on Dense, Top-$k$, and fixed-threshold Top-$p$ to isolate the central question of whether Top-$p$ routing can be made both compute-controllable and more effective than Top-$k$. Expert-choice routing (Zhou et al., 2022) relies on global token assignment and is less directly compatible with autoregressive LLM pre-training. Auxiliary zero-compute expert methods (Jin et al., 2024b) are orthogonal to DTOP-$p$ and can be combined with it in future systems.

# I. Limitations

While DTOP-$p$ demonstrates superior performance and sparsity control compared to standard Top-$k$ and fixed-threshold Top-$p$ routing, several limitations remain. First, due to computational resource constraints, our scaling investigations are limited to models with a maximum of 13.6B total parameters and training runs up to 300B tokens. While we observe consistent scaling laws within this regime, validating the effectiveness of DTOP-$p$ on frontier-scale foundation models (e.g., exceeding 100B active parameters or training on trillion-token corpora) remains future work. Second, although the PI controller proves robust to initialization in our experiments, it may require calibration when shifting to significantly different architectures or modalities outside of NLP and CV. Finally, our current implementation focuses on applying DTOP-$p$ to the router; the interaction between dynamic sparsity control and other advanced MoE techniques, such as expert-choice routing (Zhou et al., 2022) or heterogeneous expert architectures (Jin et al., 2024b) are not explored.

# J. Reproducibility Statement

We are committed to ensuring the reproducibility of our results. To this end, we provide the following resources:

- **Data:** All NLP datasets used in this work are either open-source or described with sufficient detail in Section 5.1 and Appendix C and D to allow reconstruction using comparable public data.

- **Algorithm Details:** The complete training procedure is formally described in Algorithm 1.

- **Hyperparameters:** We provide comprehensive tables of hyperparameters for all reported models. Table 1 and 5 detail the model architectures, while Table 6 in Appendix C lists the exact training configurations, including learning rates, batch sizes, optimizer settings, and loss coefficients.

- **Ablation Studies:** The specific hyperparameter ranges and configurations for our ablation studies (PI controller coefficients, loss coefficients, etc.) are detailed in the Figures in Section 5.5 and Appendix F and G to facilitate verification of our design choices.

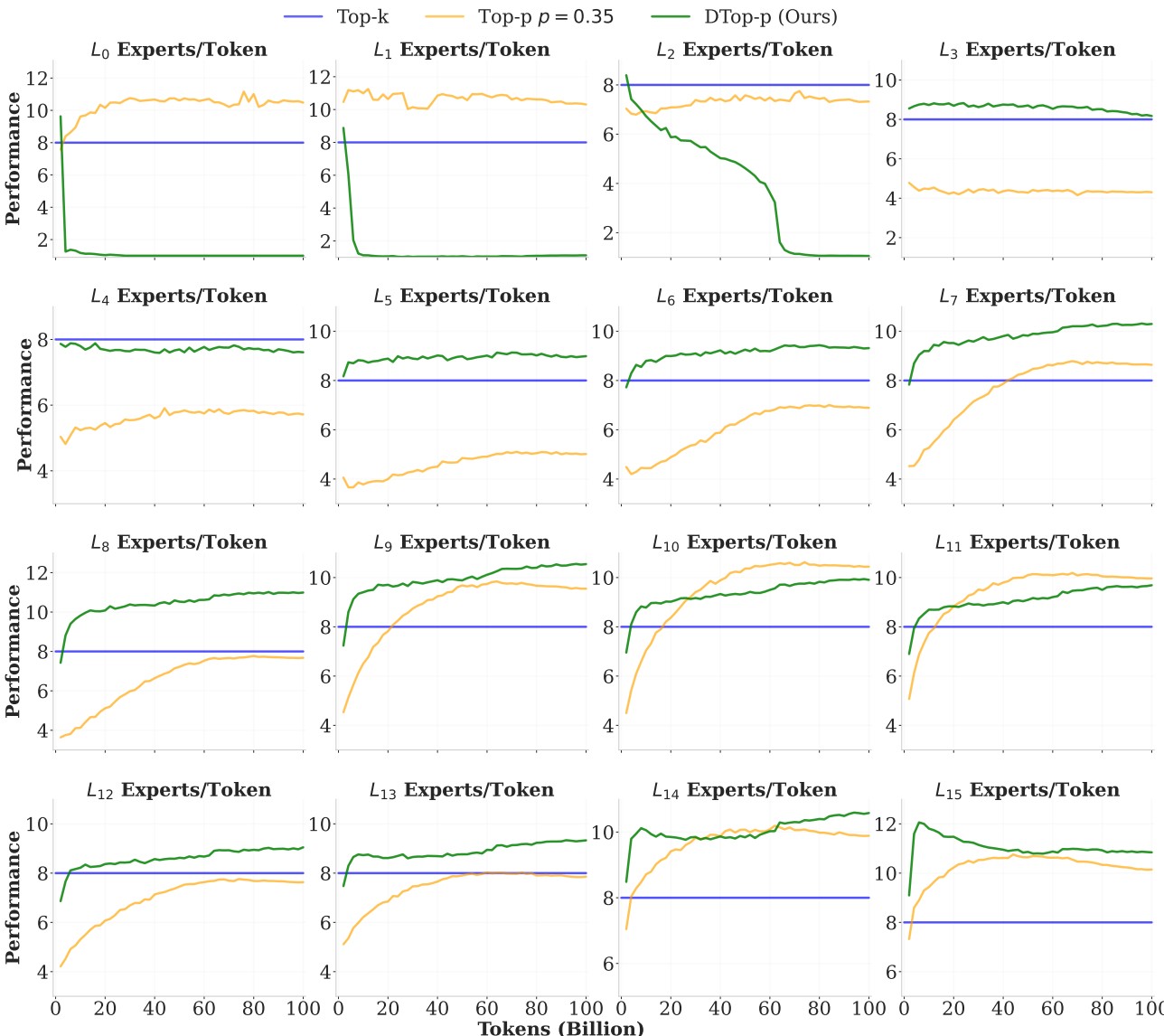

*Figure 8.* Comprehensive layer-wise evolution of activated experts per token for the MoE-1.3B-6.9B-64E8A model (100B tokens) across all 16 layers. DTOP-$p$ learns to activate fewer experts in shallow layers while utilizing more experts in deeper layers.

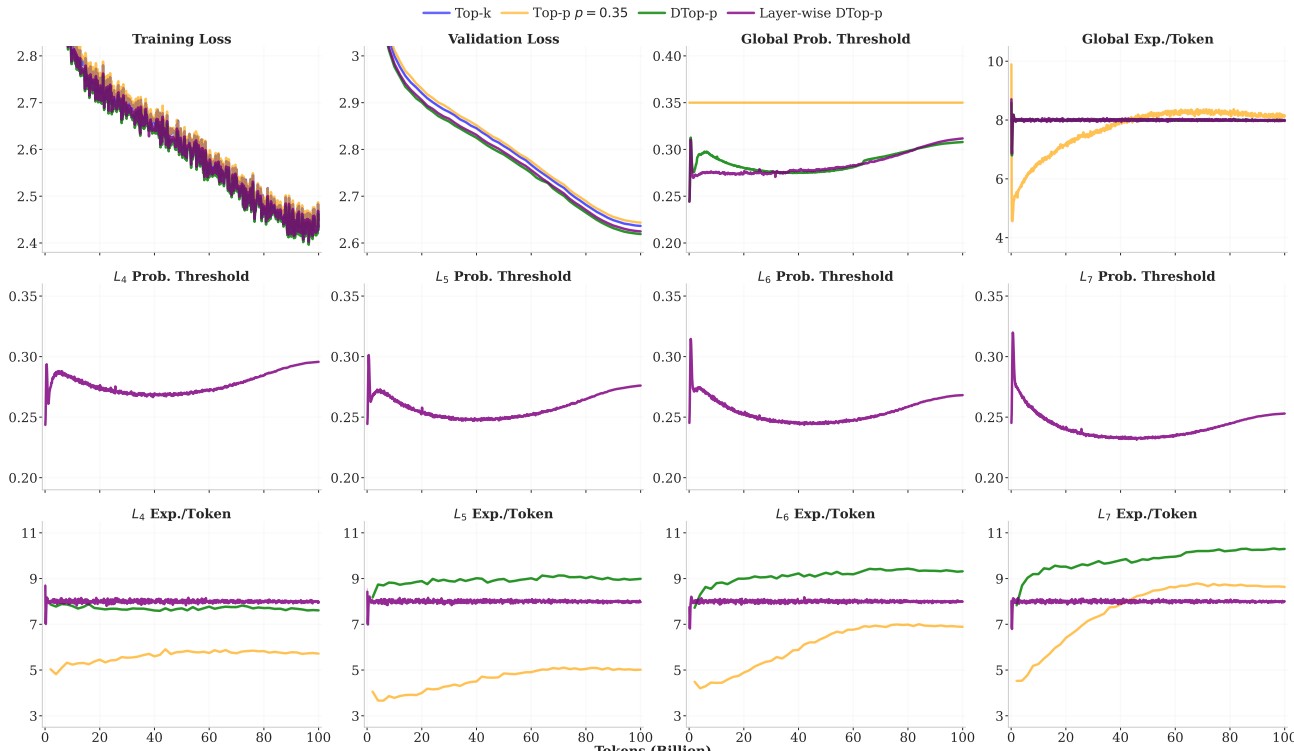

*Figure 9.* Performance comparison of Top-$k$, Top-$p$, DTOP-$p$, and layer-wise DTOP-$p$. Layer-wise DTOP-$p$ achieves near-perfect tracking of both global and layer-wise compute to the target budget by using layer-specific probability thresholds, while still outperforming Top-$k$ and Top-$p$ under the same FLOPs.

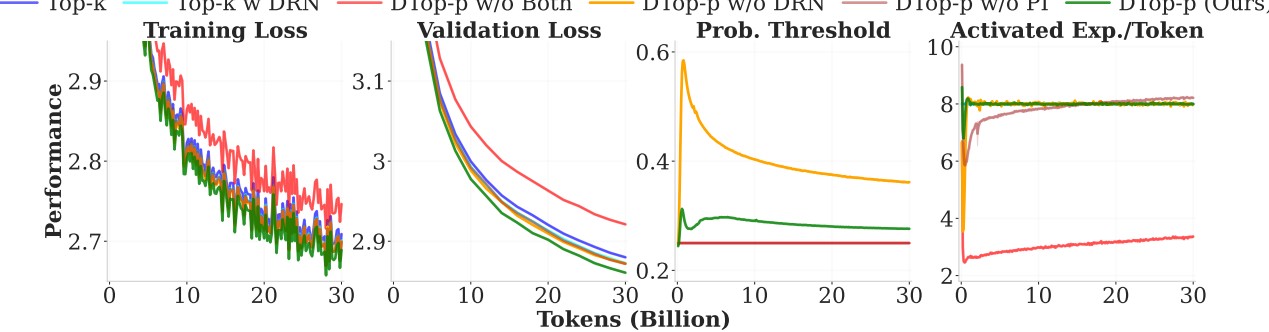

*Figure 10.* Ablation study of the PI controller (PI) and Dynamic Routing Normalization (DRN). DTOP-$p$ achieves optimal performance when both components are combined, surpassing Top-$k$ with DRN.

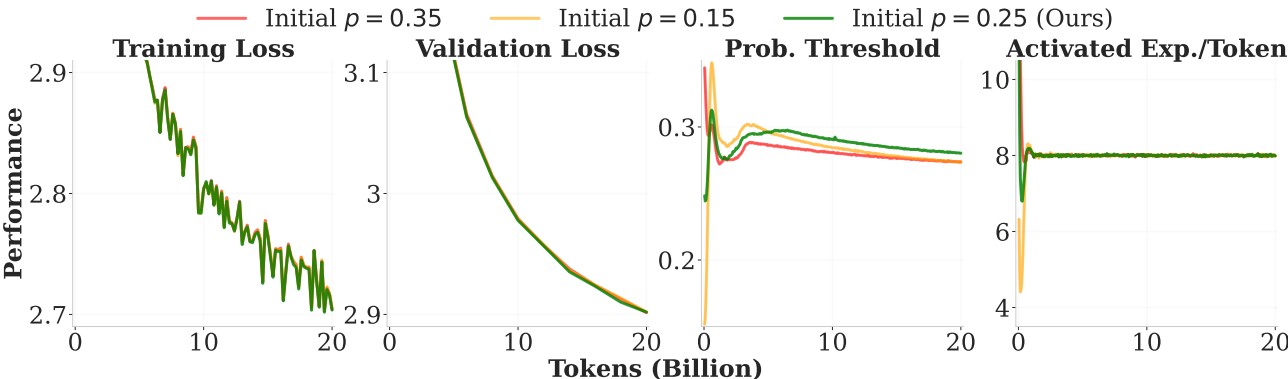

*Figure 11.* Performance of the PI controller across various probability initialization values. Unlike fixed-threshold methods, DTOP-*p* is insensitive to initialization, consistently converging to optimal performance.

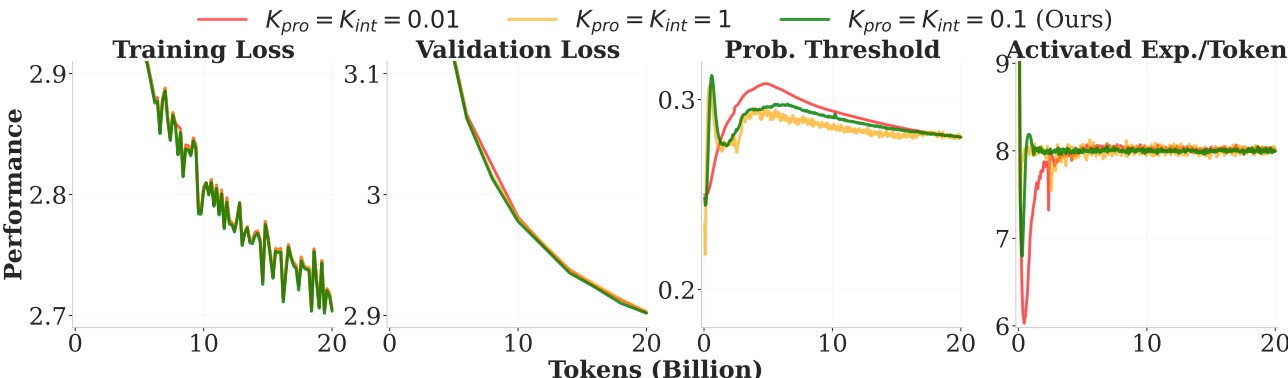

*Figure 12.* Performance of the PI controller across various $K_{pro}$ and $K_{int}$ values. DTOP-*p* achieves the best performance with moderate PI coefficient values.

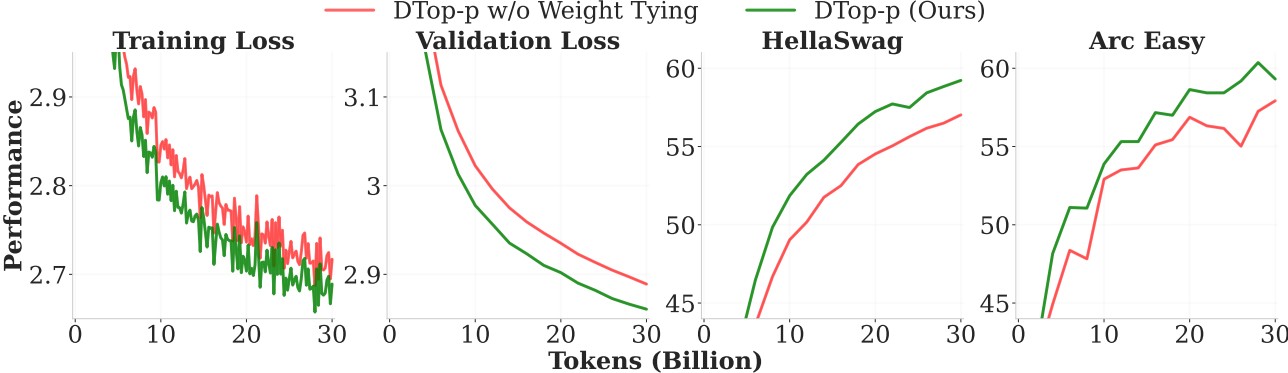

*Figure 13.* Ablation results for weight tying. The configuration with weight tying yields significantly better performance than the untied counterpart.

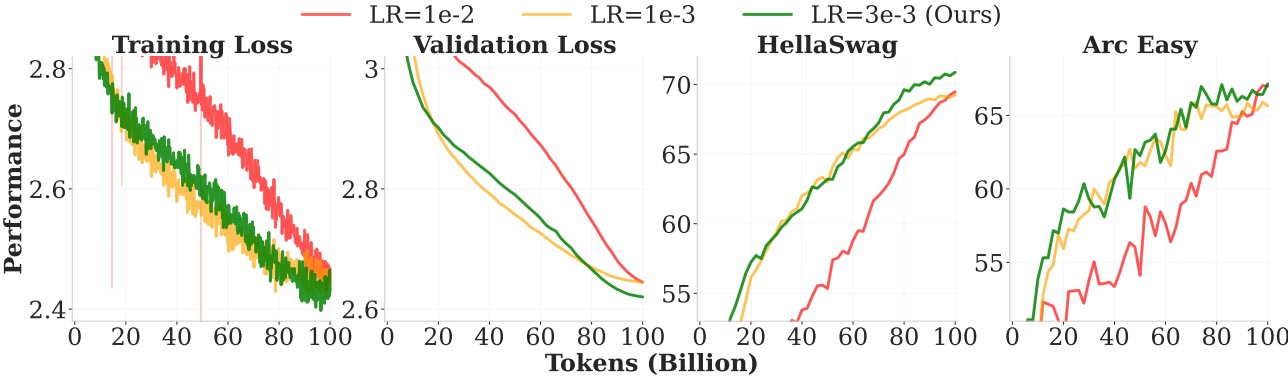

*Figure 14.* AdamW Learning Rate tuning results. A moderate Learning Rate of 3e-3 achieves the best performance, striking a balance between stability and convergence speed.

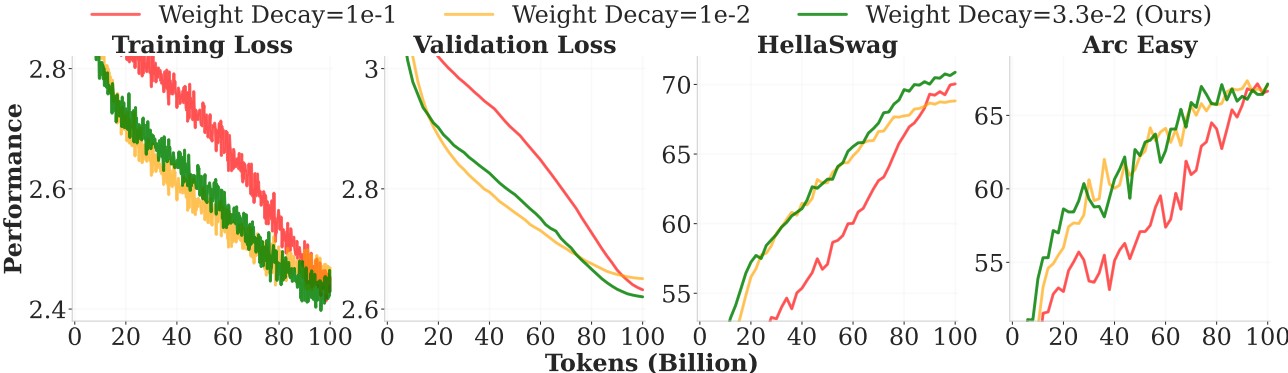

*Figure 15.* AdamW Weight Decay tuning results. A moderate Weight Decay of 3.3e-2 achieves the best performance, avoiding both over-regularization and suboptimal convergence.

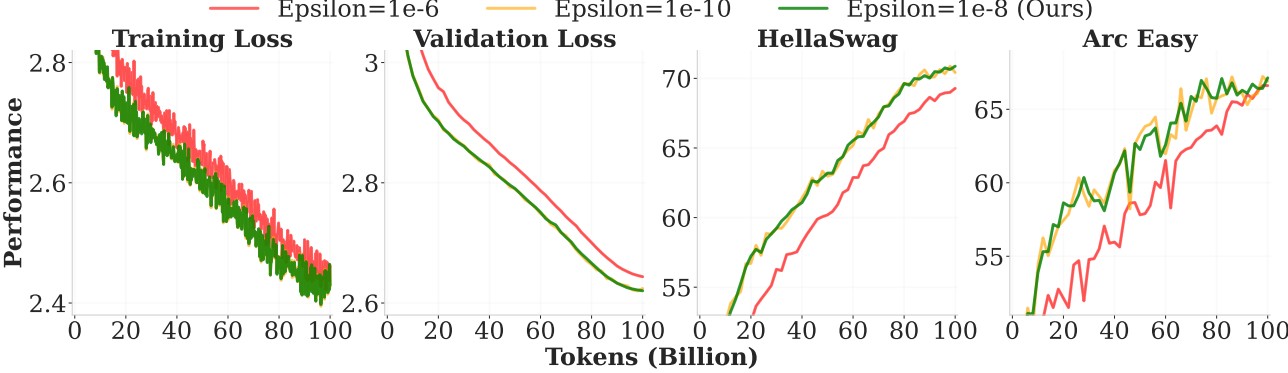

*Figure 16.* AdamW Epsilon tuning results. Smaller Epsilon values (1e-8 and 1e-10) yield superior performance compared to larger values.

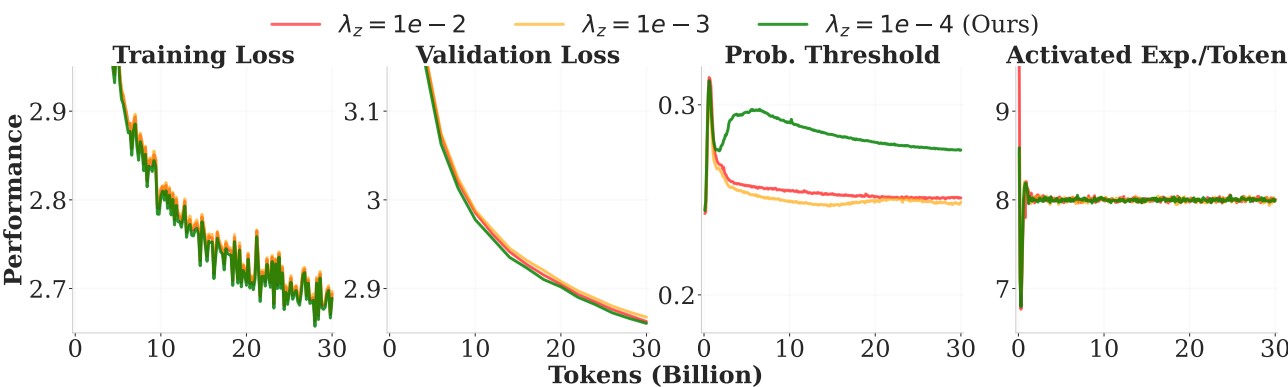

*Figure 17.* Language Modeling Z-Loss coefficient tuning results. A small coefficient of $1 \times 10^{-4}$ achieves the best performance.

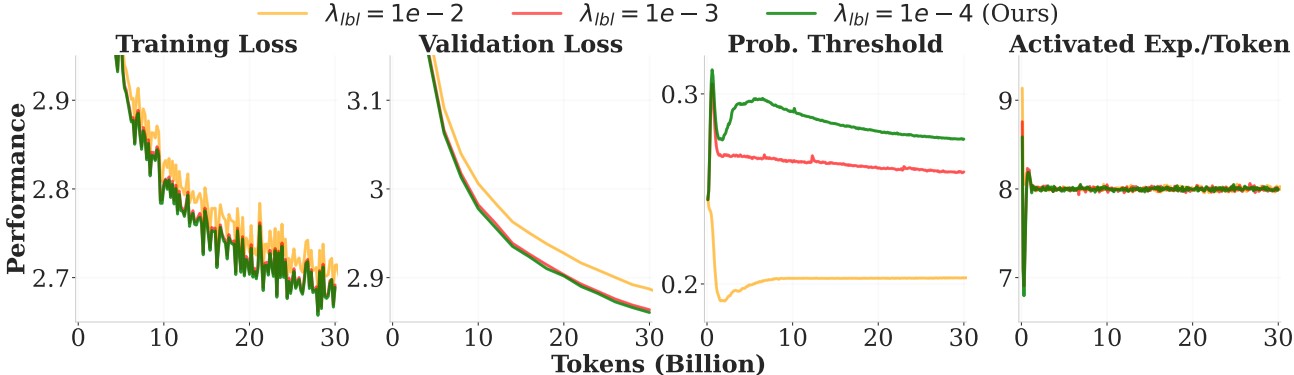

*Figure 18.* Load Balancing Loss coefficient tuning results. A small coefficient of $1 \times 10^{-4}$ achieves the best performance.

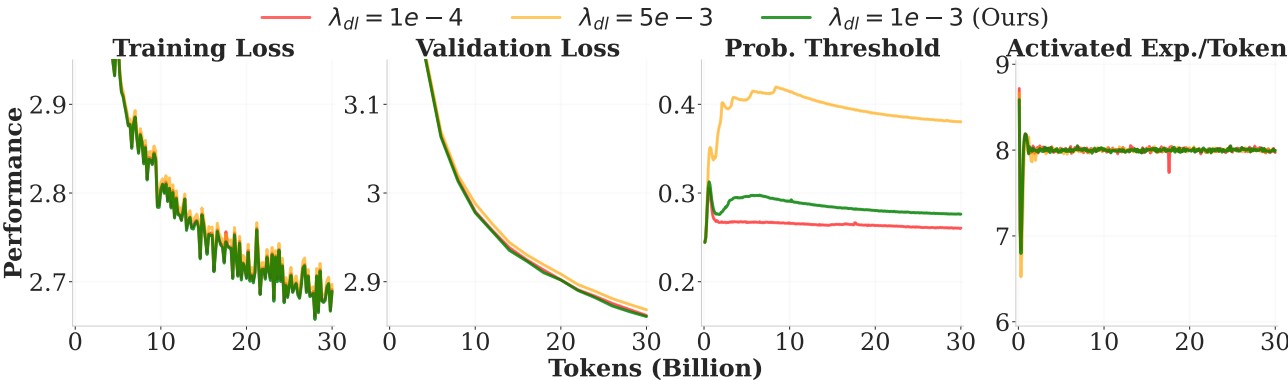

*Figure 19.* Dynamic Loss coefficient tuning results. A moderate coefficient of $1 \times 10^{-3}$ achieves the best performance for DTOP-$p$.

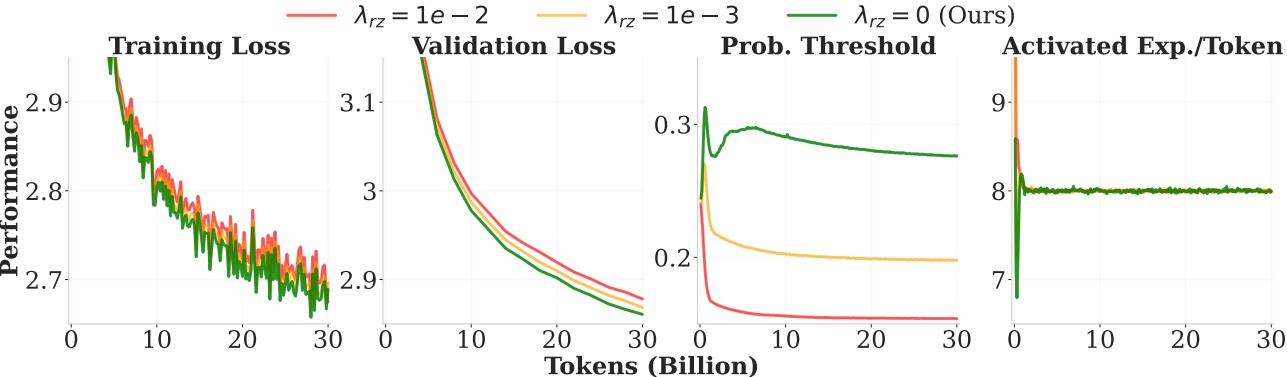

*Figure 20.* Router Z-Loss coefficient tuning results. DTOP-$p$ performs best without Router Z-Loss ($\lambda_{rz} = 0$).

