# OpenReview forum: "DTop-p MoE: Sparsity-Controlled Dynamic Top-p MoE for Foundation Model Pre-training"
_ICML.cc/2026/Conference — ICML 2026 regular_

### Official Review · Reviewer_v1yJ · 2026-03-08

**Soundness:** 3
**Presentation:** 3
**Significance:** 3
**Originality:** 3
**Overall Recommendation:** 4
**Confidence:** 5

**Summary:**

This paper addresses the issue that the fixed threshold p in traditional top-p methods leads to uncontrollable numbers of activated experts at each layer. Therefore, we propose using PI to maintain overall control over the number of activated experts. Additionally, we introduce dynamic routing normalization to achieve diversity in the distribution of experts across layers.

**Compliance With Llm Reviewing Policy:**

Affirmed.

**Final Justification:**

Most of my concerns have been solved, and based on the contribution of this paper, I prefer to keep my score.

**Key Questions For Authors:**

1.Why not learn layer-specific thresholds directly, instead of using a global PI-controlled threshold plus layer-wise logit scaling?
2.The PI gains are manually tuned rather than learned, which may limit portability across architectures or scales. Have you tried some dynamic methods?
3.How strong is the external validity of the empirical results? why not directly replace some successfully sota moe models and replace their top-k method with dtop-p, such as qwen30b-moe, olmoe, or deepseek-moe.
4.The paper seems to combine a global PI controller that enforces a target average number of activated experts with a layer-wise normalization mechanism that deliberately induces heterogeneous sparsity patterns across layers. It is therefore unclear whether these two components are fully aligned in objective. In particular, the paper does not clearly explain whether the learned \theta_l values are regularized for load balancing or are driven purely by the task loss, nor does it analyze whether layer-wise expert allocation becomes overly skewed under the global budget control.

**Limitations:**

Yes.

**Strengths And Weaknesses:**

Soundness. The paper is technically solid overall and the core idea is well motivated by the instability of fixed-threshold Top-p routing shown in the preliminary study. A weakness is that the method is justified almost entirely empirically, and the control-theoretic component is used in a practical rather than principled way.

Presentation. The paper is generally clear and well structured, and the progression from the failure mode of fixed Top-p to the DTop-p design is easy to follow. The abstract and introduction lack the necessary explanation of top-k and top-p, particularly why top-p is preferable to top-k.

Significance. The paper addresses how to obtain adaptive routing without losing control over compute budgets, though the demonstrated gains are incremental rather than transformative.

Originality. The paper’s main novelty lies in combining Top-p routing with a PI-controller-based threshold adaptation and layer-wise routing normalization under a global sparsity budget. This is not a fundamentally new routing paradigm, but it is a thoughtful and reasonably original combination that addresses a real weakness of existing Top-k and fixed Top-p methods.

---

> ### Author Rebuttal · Authors · 2026-03-29
>
> Thank you for the constructive review.
>
> > W1 Control-theoretic component is more practical than principled
>
> We agree that our evidence is primarily empirical. Our goal is not a full control-theoretic treatment, but a feedback-control mechanism that is more principled than heuristic Top-p. As discussed in Section 3, Top-p suffers from uncontrolled global compute, sensitivity to the initial threshold, costly tuning, and only marginal gains over Top-k. To address this while preserving its flexibility, we introduce a PI controller that updates the threshold from the sparsity error to enforce adaptive budget control. This avoids differentiating through the non-differentiable threshold, and the integral term reduces steady-state bias, leading to more stable global budget control.
>
> > W2 Why Top-p is preferable to Top-k
>
> Top-k imposes a uniform sparsity pattern across all tokens and layers, ignoring that tokens vary in difficulty and layers vary in computational needs. In contrast, Top-p routing adapts the number of experts to router confidence: high-confidence tokens use fewer experts, while more ambiguous tokens recruit more. This token-adaptive and layer-adaptive behavior makes Top-p a more expressive routing paradigm for enhanced reasoning.
>
> > W3 Gains are incremental
>
> We agree that the gains are incremental, but they are meaningful because they are achieved without increasing average FLOPs, while also providing precise control over dynamic compute allocation. This is also consistent with prior MoE routing works [1,2], which report gains of similar scale over Top-k. Empirically, on NLP, DTop-p improves the average score by about 2% across 13 downstream benchmarks, and it also consistently outperforms Top-k on CV. Moreover, this advantage grows across scaling regimes, suggesting that the benefit of DTop-p remains practically meaningful as training scales.
>
> [1]Skywork-MoE: A Deep Dive into Training Techniques for Mixture-of-Experts Language Models
>
> [2]ReMoE: Fully Differentiable Mixture-of-Experts with ReLU Routing
>
> > Q1 Learn layer-specific thresholds
>
> Our design cleanly separates how many experts to use globally from how to distribute them across layers. This modularity is a strength: the PI controller provides hard budget guarantees, while DRN provides soft, data-driven layer specialization. Learning separate thresholds for each layer would impose stronger constraints on layer-wise routing and reduce the ability to reallocate compute across layers, which can hurt performance. We provide an ablation of layer-wise threshold control in **Figure:[link](https://postimg.cc/WdCBSSb2)**. It achieves near-perfect control of the per-layer compute and still outperforms Top-k, but performs slightly worse than global DTop-p. This suggests that stricter per-layer control reduces useful cross-layer flexibility.
>
> > Q2 PI gains are manually tuned
>
> In NLP and CV experiments, DTop-p remains robust across modalities, model sizes, and dataset sizes, using the same PI gains throughout. The PI ablations also show robustness to different probability initializations and rapid threshold adaptation. This suggests that the PI gains act as a small set of stable control hyperparameters, not brittle tuning knobs. Moreover, DTop-p is already more portable than Top-p. We will clarify this empirical robustness in the revision and note that automated gain adaptation is a promising future direction.
>
> > Q3 External validity/replace routing in SOTA MoE models
>
> The significance of our gains is addressed in W3. Our goal is to evaluate DTop-p under controlled pre-training settings, rather than on top of highly customized released systems whose behavior depends on many additional engineering choices. For this reason, we build the MoE modules for LLM and DiT from scratch in our internal codebase instead of using Qwen30B-MoE. DTop-p is largely orthogonal to many existing MoE training tricks in OLMoE and DeepSeek-MoE, and our ablations suggest that some of these tricks may interact differently with DTop-p and Top-k, such as DRN and Router Z-Loss (Figures 9 and 19).
>
> > Q4 Whether PI and DRN are aligned and $\theta_l$ is regularized for LB
>
> These two components are complementary: the PI controller constrains the batch-level number of activated experts, while DRN reshapes each layer’s routing distribution under the global budget. Together, they enable adaptive compute allocation across tokens and layers, which is also supported by Figure 7, where PI + DRN gives the largest improvement over Top-k.
>
> $\theta_l$ is learned jointly with the model parameters through the pre-training loss, without an explicit regularizer. In practice, this does not lead to collapse: Figure 5 shows a meaningful layer-wise pattern rather than pathological skew, and the retained LB loss helps prevent unhealthy expert imbalance. We also find that DRN brings larger gains in DTop-p than in Top-k (Figure 9). Our framework can also be extended to layer-wise control as discussed in Q1.

---

> > ### Author Rebuttal · Reviewer_v1yJ · 2026-04-01
> >
> > Most of my concerns have been solved, and based on the contribution of this paper, I prefer to keep my score.

---

> > > ### Author Response · Authors · 2026-04-01
> > >
> > > We sincerely thank the reviewer for replying and acknowledging that most concerns have been adequately addressed. We are happy to further discuss any remaining questions to further improve the quality of our work.
> > >
> > > We also appreciate the opportunity to further clarify the paper’s contribution:
> > >
> > > - **Comprehensive analysis of existing Top-$p$ MoE.** In both NLP and CV (Figures 2/3/4), we demonstrate that fixed-threshold Top-$p$ MoE suffers from uncontrolled computational costs, high sensitivity to the initial threshold, expensive tuning to identify an optimal threshold under a fixed compute budget, and only marginal gains over Top-$k$ when computation is matched. These are fundamental obstacles for large-scale pre-training, where computational budgets must be strictly controlled. DTop-$p$ is proposed to directly address this open problem.
> > >
> > > - **PI-controlled dynamic threshold for near-perfect budget control.** The PI controller, grounded in decades of classical control theory, provides near-perfect global and layer-wise budget control during both training and inference, thereby enhancing stability and performance (Figures 6/7 and **Figure:[link](https://postimg.cc/WdCBSSb2)**). This principled feedback mechanism for dynamically learning the probability threshold is fundamentally different from simply tuning a fixed hyperparameter for the entire pre-training process — it addresses the non-differentiability of the threshold through a well-established control-theoretic framework.
> > >
> > > - **Dynamic Routing Normalization (DRN) for layer-wise adaptivity.** DRN introduces learnable, layer-specific normalization coefficients that enable each layer to more independently modulate its expert selection, enhancing cross-layer expressiveness and accommodating the varying computational needs of layers with different levels of specialization (Figures 5/7/8 and **Figure:[link](https://postimg.cc/WdCBSSb2)**). This decouples layer-wise routing behavior from the global threshold — a capability absent from prior Top-$p$ and Top-$k$ methods.
> > >
> > > - **Comprehensive empirical and practical investigations.** DTop-$p$ achieves better performance over both Top-$k$ and Top-$p$ MoE while near-perfectly maintaining the same global compute budget as Top-$k$. Our evaluation spans: (1) multiple modalities (NLP and CV), demonstrating generality; (2) four scaling regimes (expert granularity, expert capacity, model size, and dataset size), providing strong evidence for scalability — which, to our knowledge, has not been systematically studied for Top-$p$ routing before; and (3) 13 downstream tasks covering commonsense reasoning, language understanding, reading comprehension, world knowledge, and symbolic problem solving. Beyond main results, we provide thorough investigations including detailed ablations of the PI controller and DRN (validating near-perfect budget control, stability, and genuinely adaptive routing), loss function designs (language modeling loss with Z-Loss, load balancing loss, dynamic loss, and router Z-Loss), AdamW optimizer tuning (learning rate, weight decay, epsilon), and architectural choices such as weight tying — offering practical guidance for DTop-$p$ usage and tuning.
> > >
> > > Together, these contributions demonstrate DTop-$p$'s clear motivation, strong practical impact, and validate the effectiveness of integrating control-theoretic techniques and dynamic normalization into MoE architectures for future research.
> > >
> > > We thank the reviewer again for the constructive engagement and would be very happy to clarify and address any further concerns.

---

### Official Review · Reviewer_993r · 2026-03-11

**Soundness:** 4
**Presentation:** 4
**Significance:** 3
**Originality:** 3
**Overall Recommendation:** 4
**Confidence:** 5

**Summary:**

This paper addresses the rigid limitations of standard Top-k routing in MoE and the uncontrollable computational burden of conventional Top-p routing by proposing DTop-p MoE.

The core innovation of this method lies in introducing a PI controller from classical control theory to dynamically adjust the Top-p threshold, thus preserving the flexibility of Top-p routing while meeting a preset computational budget. Furthermore, the paper introduces DRN, allowing different network layers to learn differentiated expert activation patterns under a global threshold.

Experiments covering LLM and DiT demonstrate that this method outperforms existing Top-k and fixed-threshold Top-p baselines in both performance and scaling.

**Compliance With Llm Reviewing Policy:**

Affirmed.

**Key Questions For Authors:**

1. Do you maintain consistency in the use of $K_{pro}$ and $K_{int}$ across models of different sizes?

2. Dynamic routing tends to assign more work to certain experts. Does this conflict with load balancing loss? How do you balance these two?

3. During inference, are there significant differences in the range of computational fluctuations caused by different prompts?

**Limitations:**

Yes

**Strengths And Weaknesses:**

Strengths

1. The non-differentiability of the Top-p threshold is solved using a PI controller, avoiding complex gradient estimation.
2. Heuristic hierarchical behavior is observed: The model learns to reduce expert activation in shallow layers (handling general features) and increase expert activation in deeper layers (handling complex semantics).
3. The method not only performs excellently on LLM but also successfully demonstrates its generality on DiT.

Weaknesses
1. Although FLOPs are matched, no comparative results are provided for the actual training/inference speed of this method, especially in expert-parallel settings.
2. It is questionable whether this method, which balances the sparsity of all layers based on a "global threshold," will still be effective on larger-sized models.

---

> ### Author Rebuttal · Authors · 2026-03-28
>
> Thank you for the thoughtful and positive review. Below, we address the concerns in detail.
>
> > W1: Actual speed versus matched FLOPs
>
> We thank the reviewer for the suggestion. We compare FLOPs, training, and inference speed between Top-k and DTop-p in Table T2. Both methods use the same architecture, and DTop-p activates 7.99 experts/token on average versus Top-k’s fixed 8, yielding essentially identical FLOPs.
>
> This is also reflected in runtime: training takes 1.63 s/step for both methods, with throughput of 646 Ktok/s for Top-k and 645 Ktok/s for DTop-p on 32× H100 GPUs with FSDP. During inference, both methods show nearly identical prefill latency/throughput; in autoregressive decoding at batch size 1 on one H100 GPU, both achieve about 320 tok/s with 3.08 ms/token latency.
>
> Overall, DTop-p adds negligible runtime overhead relative to Top-k while delivering the quality improvements reported in the main paper, showing that the matched-FLOPs comparison is also reflected in actual training and inference speed.
>
> Table T2: Training and inference FLOPs and Speed comparison between Top-k and DTop-p.
> Metric | Top-k | DTop-p (Ours)
> -|-|-
> Total / Activated Params | 6.92B / 1.28B | 6.92B / 1.28B
> Avg Experts/tok | 8.00 | 7.99
> Forward FLOPs/tok | 2.49 G | 2.49 G
> Decode FLOPs/tok (ctx=2048) | 2.63 G | 2.63 G
> Training Step Time (s) | 1.63 | 1.63
> Training Throughput (Ktok/s) | 646 |645
> Prefill Latency (ms/tok, seq=2048) | 26.9 | 26.9
> Prefill Throughput (Ktok/s) | 76.4 | 76.3
> Decode Latency (ms/tok, ctx=2048) | 3.08 | 3.08
> Decode Throughput (tok/s) | 327 | 328
>
> > W2: Whether the global-threshold method will remain effective at larger scales
>
> We would like to clarify that DTop-p does not rely on a raw global threshold alone. The PI controller governs the global average compute budget, while DRN explicitly decouples this global threshold from layer-local logit statistics through a learnable layer-specific scale. This design allows the model to maintain a global compute target while still learning distinct sparsity patterns across depths, which in turn leads to improved performance over Top-k MoE (Figure 7). Empirically, our scaling studies show that DTop-p consistently widens its advantage as expert granularity, expert capacity, and model size increase, including up to 2.4B active / 13.6B total parameters (Section 5.3). We agree that validation on even larger frontier-scale MoE models would further strengthen this claim, but the current evidence supports the scalability of the design within the evaluated range.
>
> > Q1: Consistency of PI coefficients across model sizes
>
> As stated in Appendix C, our scaling studies use the main hyperparameter setting unless otherwise specified; therefore, we use the same PI coefficients across model sizes. This consistency is important because it shows that DTop-p does not rely on model-size-specific retuning of the controller.
>
> In addition, the PI-controller tuning results show that DTop-p is robust to probability initialization and that moderate PI coefficients provide the best trade-off between convergence speed and stability, while excessively large values introduce instability and very small values lead to slow convergence. Taken together, these results suggest that the controller is not overly sensitive to precise coefficient choices within a reasonable range. We will clarify this hyperparameter consistency more explicitly in the revision, especially in Section 5.1.
>
> > Q2: Interaction between dynamic routing and load balancing (LB) loss
>
> These two components are designed to address different levels of the routing problem and are therefore complementary rather than conflicting. Dynamic routing adaptively allocates computation across tokens and layers, while the LB loss regularizes the utilization of experts to avoid router collapse and overly skewed expert assignment.
>
> Accordingly, we retain the LB loss in the final objective to ensure that the flexibility introduced by dynamic routing does not come at the cost of unhealthy expert imbalance. In this sense, DTop-p enables adaptive token-/layer-level compute allocation while still enforcing regularized expert-level utilization.
>
> > Q3: Compute fluctuation across prompts during inference
>
> DTop-p closely tracks the target activation budget on both training and validation data, and shows a moderate and stable standard deviation in activated experts per token, whereas fixed-threshold Top-p exhibits much larger variance (Figure 6). This already suggests more stable routing behavior and better control of computation.
>
> We agree that a dedicated prompt-conditioned inference analysis would make this claim more complete. In the revision, we will include additional analyses of activated experts at the token and task levels, with both layer-wise and global-wise views, to better characterize downstream routing patterns and inference-time budget fluctuations.

---

> > ### Author Rebuttal · Reviewer_993r · 2026-04-07
> >
> > I thank the authors for their detailed responses. All my concerns have been addressed, and I will maintain my score.

---

> > > ### Author Response · Authors · 2026-04-07
> > >
> > > We sincerely thank Reviewer 993r for the constructive feedback throughout the review process and for the time and care devoted to evaluating our rebuttal. We are very glad that our responses have addressed all of the reviewer’s concerns.
> > >
> > > In the final version, we will incorporate the clarifications from the rebuttal and add further analysis of activated experts at both the token and task levels, from both layer-wise and global perspectives, to better characterize downstream routing behavior and inference-time budget variation.
> > >
> > > Thank you again for your thoughtful and positive engagement, which has helped us improve the paper.

---

### Official Review · Reviewer_oJd6 · 2026-03-12

**Soundness:** 3
**Presentation:** 3
**Significance:** 3
**Originality:** 3
**Overall Recommendation:** 4
**Confidence:** 3

**Summary:**

In this paper, the authors propose DTop-p MoE to improve the original Top-p MoE models. Specifically, it designs a dynamic threshold based on proportional–integral control to dynamically control the total number of active experts across the whole model, allowing the number of active experts to remain stable at the beginning of training and improving training stability. Furthermore, dynamic routing normalization is designed to give each layer more flexibility in expert selection. The authors evaluate the proposed MoE on both NLP and CV tasks, which shows improved results over the baselines.

**Compliance With Llm Reviewing Policy:**

Affirmed.

**Final Justification:**

The author provides a detailed explanation of the method, motivation, empirical performance, and computational cost, which addresses most of my concerns. I would like to raise to a positive score.

**Key Questions For Authors:**

1. Could the authors provide the exact numbers for both pre-training loss and validation loss for the NLP and CV tasks? Downstream performance can vary, but the loss is a more direct signal of model performance.

2. Could the authors explain why the proposed proportional–integral control helps to avoid differentiation of the top-p threshold? It seems to me that this is just a threshold hyper-parameter tuning during the training. The model still cannot learn a best threshold based on data.

**Limitations:**

See above.

**Strengths And Weaknesses:**

## Strengths

1. The paper writing is clear and easy to follow.
2. The authors provide extensive experiments and ablation studies to validate the proposed method.

## Weaknesses

1. The novelty of the paper is limited. It appears more like an engineering trick for Top-p MoE. The authors did not fully explain why a dynamic threshold or dynamic routing normalization can lead to better performance than top-p and top-k MoE.

2. The performance gain over the baseline is marginal. In particular, in Figure 3, the validation loss improvement over Top-k is only about 0.01 (I cannot infer the exact number from the figure), which is quite small. This raises the question of why one should not simply use the standard Top-k MoE.

3. Although the global activation is more stable compared to the original Top-p MoE, the per-layer activation difference is large (2 in shallow layers and 11 in deeper layers, based on Figure 5). This is still unpredictable and uncontrollable before training, which may pose challenges for deployment (since deeper layers require more resources and bandwidth).

4. Furthermore, the discrepancy in the number of active experts across layers also raises concerns about model convergence. Specifically, it is common for MoE models to converge faster in the early phase by leveraging fewer experts (a small number of experts become well-trained while others remain under-trained), but this can eventually become suboptimal compared to the baseline Top-k. Training on a larger data scale (e.g., >400B tokens) would be helpful to rule out this factor. I am also not asking the author to provide these results due to possible resource and time constraints. But any ablation that can rule out this factor would be appreciated.

---

> ### Author Rebuttal · Authors · 2026-03-29
>
> Thank you for the constructive review.
>
> > W1 & Q2 Limited novelty; why PI and DRN help; why PI avoid differentiation
>
> We respectfully disagree that DTop-p is merely an engineering trick. The core contribution is a principled formulation of a previously intractable problem. As discussed in Section 3, we identify that Top-p has three key limitations: uncontrolled global compute, sensitive and expensive threshold tuning, and only marginal gains over Top-k. To address these issues while preserving Top-p’s flexibility, we introduce a novel PI controller to adapt the threshold from the sparsity error, enabling stable global budget control and reducing steady-state bias. We also introduce DRN to reshape layer-wise routing distributions under the global budget (Section 4.2).
>
> Sections 3 and 5.5 explain why Top-p brings limited gains, and why PI + DRN improves performance. In Top-p, the global activated experts vary substantially during training. As shown in Figures 2 and 7, Top-p and DTop-p without PI spend a long stage of pre-training far below the target activation level, which hurt performance. PI keeps the global activation near the target while allowing adaptive compute allocation across tokens of different difficulty [1]. DRN makes routing more discriminative and adaptive across layers. This is consistent with prior findings [2,3] that overly flat expert distributions can hurt performance and that different layers may require different amounts of computation. The ablations in Section 5.5 show that DTop-p performs best with PI + DRN.
>
> Regarding differentiation, our claim is not that the threshold becomes differentiable. Rather, PI avoids differentiating through the threshold by updating it between training steps from the observed sparsity error. This is important because the threshold determines the router mask and is inherently non-differentiable inside routing. Empirically, PI achieves near-perfect tracking of activated experts during training and validation (Figures 4 and 6). Therefore, the threshold updated by PI can be viewed as the “optimal” for tracking the target computation. Moreover, DRN is learned via gradient descent and is fully data-driven. The combination ensures the model learns how to better distribute experts while maintaining how many experts to use globally.
>
> [1]Harder task needs more experts: Dynamic routing in MoE models
>
> [2]Skywork-MoE: A Deep Dive into Training Techniques for Mixture-of-Experts Language Models
>
> [3]Scaling Vision with Sparse Mixture of Experts
>
> > W2 & Q1 Performance gain appears small; exact losses
>
> We report the final training and validation losses from Figures 3 and 4 in **Table:[link](https://postimg.cc/8JmB7zYj)**. Although the absolute loss gap is small, it is meaningful in pre-training, especially under the same FLOPs budget. In our experiments, it translates into an average of about 2\% downstream improvement across 13 benchmarks. We note that Top-k is a strong baseline as it is carefully tuned, making the comparison conservative and fair. Similar-scale gains over Top-k have also been reported in prior MoE works such as [2,4]. Moreover, the advantage over Top-k increases as model size, expert granularity, and expert capacity scale (Section 5.3).
>
> [4]ReMoE: Fully Differentiable Mixture-of-Experts with ReLU Routing
>
> > W3 Uneven per-layer activation may hinder deployment
>
> From a deployment perspective, the latency of an MoE layer is approximately linear in the number of activated experts: layers with fewer activated experts run faster, while layers with more activated experts run slower. However, because DTop-p matches the average activated experts of Top-k, the overall resource usage, end-to-end latency, and memory remain similar.
>
> From a modeling perspective, we view this variation as a feature rather than an instability. DRN allows different layers to learn different sparsity levels adaptively, consistent with prior findings that different layers exhibit different specialization patterns [3]. Figure 7 further shows that DRN improves performance, and we retain the LB loss to prevent expert-level collapse.
>
> If identical activation per layer is required, our framework can be readily extended to layer-wise threshold control. In this variant, different layers each follow their own target compute budget, which provides much tighter per-layer control, although it may reduce the freedom to dynamically reallocate compute across layers. As shown in **Figure:[link](https://postimg.cc/WdCBSSb2)**, layer-wise DTop-p achieves near-perfect per-layer budget control while still outperforming Top-k.
>
> > W4 The learned sparsity pattern may help early convergence, but become suboptimal later
>
> While we do not yet have experiments beyond 400B tokens, our scaling studies in Section 5.3 already cover dataset sizes up to 300B tokens (accurately 314.6B), where DTop-p consistently maintains its advantage over Top-k. We therefore view the current results as supportive evidence of scalability.

---

> > ### Author Rebuttal · Reviewer_oJd6 · 2026-04-03
> >
> > I appreciate the author's effort in addressing my concerns. Most of them are resolved.
> >
> > Although I think the key issue with unbalanced experts is that it can introduce instability in large-scale deployment (some experts receive large requests that are out of their capacity, while others are idle), which hinders their application in large-scale concurrent deployment. But the author's explanation and additional experiments also make sense and are appreciated.
> >
> > Overall, I will increase my score to weak accept.

---

> > > ### Author Response · Authors · 2026-04-03
> > >
> > > We sincerely thank the reviewer for the constructive engagement throughout the review process and for increasing the score to weak accept. We greatly appreciate the recognition of our efforts in addressing the concerns. Regarding the important point about **load imbalance and deployment stability at scale**, we would like to offer further clarification: DTop-$p$ is designed to control the compute budget on top of Top-$p$ routing, and load balance is maintained per layer through lb loss, we have totally three complementary mechanisms:
> > >
> > > **(1) Near-perfect global expert budget control.** While DTop-$p$ allows each token to adaptively select a varying number of experts, the global compute budget is precisely controlled by the PI controller, which converges to the target activation level stably and rapidly at the beginning of pre-training (Figures 3/4/6). This ensures that the average number of activated experts per token is tightly controlled and predictable — a prerequisite for stable resource allocation in deployment that fixed Top-$p$ MoE does not provide.
> > >
> > > **(2) Load Balancing Loss for balanced compute distribution among experts within a layer.** In each MoE layer, we apply the Load Balancing Loss (Equation 9, Appendix G) to encourage a more uniform distribution of tokens across all experts [5,6]. With this loss active during training, the load across different experts within each layer is balanced at deployment time, effectively preventing expert collapse [5,6].
> > >
> > > **(3) (Optional) Layer-wise budget control for near-perfect per-layer compute distribution.** When utilizing the layer-wise PI control introduced in the rebuttal for W3, the compute budget for each layer is controlled to near-perfectly match its target level. Layer-wise DTop-$p$ can further encourange large scale deployment. Importantly, layer-wise DTop-$p$ still enables dynamic token-level compute allocation and outperforms Top-$k$ MoE.
> > >
> > > With the combination of (1)+(2)+(3), the global compute budget is near-perfectly controlled, while the layer-level and expert-level loads are also balanced — avoiding skewed compute distribution while still enabling adaptive, dynamic token-wise and layer-wise allocation for enhanced reasoning.
> > >
> > > We will include the experiments on layer-wise DTop-$p$ in **Figure: [link](https://postimg.cc/WdCBSSb2)** and add a dedicated discussion of load balancing considerations for DTop-$p$ deployment in the revision. Thank you again for the valuable feedback, which has helped strengthen the practical perspective of our work.
> > >
> > >
> > >
> > > [5] Outrageously large neural networks: The sparsely-gated mixture-of-experts layer
> > >
> > > [6] Gshard: Scaling giant models with conditional computation and automatic sharding

---

### Official Review · Reviewer_8kdc · 2026-03-13

**Soundness:** 3
**Presentation:** 3
**Significance:** 3
**Originality:** 3
**Overall Recommendation:** 4
**Confidence:** 3

**Summary:**

The paper introduces "DTop-p MoE," a novel routing mechanism designed to improve the efficiency and flexibility of sparse Mixture-of-Experts (MoE) architectures in foundation models. The authors identify that standard Top-k routing is too rigid, while naive Top-p routing suffers from unpredictable computational costs and sensitivity to threshold hyperparameters. To solve this, DTop-p employs a Proportional-Integral (PI) controller to dynamically adjust the global probability threshold, ensuring the running sparsity matches a predefined computational budget. Additionally, it introduces Dynamic Routing Normalization (DRN) to adaptively rescale layer-specific logits, allowing for distinct routing patterns at different depths of the network. The authors demonstrate that DTop-p outperforms standard Top-k and fixed Top-p baselines across both Natural Language Processing (NLP) and Computer Vision (CV) tasks while strictly maintaining target compute budgets.

**Compliance With Llm Reviewing Policy:**

Affirmed.

**Final Justification:**

This is a strong work, supported by comprehensive experiments. While further large-scale validation would help demonstrate the approach's practical scalability, I recognize that such evaluations may be beyond the scope of typical academic computational resources. Consequently, I maintain my initial positive assessment.

**Key Questions For Authors:**

- In heavily distributed training setups (e.g., spanning hundreds of GPUs with Expert Parallelism), how is the global average of activated experts ($a_t$​) synchronized efficiently to update the threshold ($p_{t+1}$​) without introducing a synchronization bottleneck?

- How does the integral term of the PI controller handle sudden, severe data distribution shifts during training? Is there a risk of integral windup if a particularly complex batch of tokens temporarily spikes the required expert capacity?

**Limitations:**

- The approach has not yet been validated on frontier-scale foundation models (e.g., models exceeding 100B active parameters or trained on multi-trillion token corpora) due to compute constraints.

- The PI controller, while robust to initialization in the tested domains, may require significant recalibration when adapted to entirely new architectures or modalities outside of standard NLP/CV transformers.

**Strengths And Weaknesses:**

**Strengths**

- Well-Motivated Problem: The paper addresses a highly practical challenge in scaling foundation models. Unpredictable computational graphs caused by standard Top-p routing are a massive hurdle for large-scale pre-training. Solving this while maintaining dynamic routing benefits is a strong contribution.

- Novel Application of Control Theory: Using a PI controller to handle the non-differentiable nature of the Top-p threshold is a lightweight, and effective solution.

- Comprehensive Evaluation: The empirical validation is thorough. The authors test the method not only on LLMs (up to 2.4B active parameters) but also successfully apply it to Diffusion Transformers (DiTs) in the visual domain.

- Strong Scaling Analysis: The paper rigorously tests the method across different scaling axes, including expert granularity, total expert capacity, model size, and dataset size (up to 300B tokens).

**Weaknesses**

- System-Level Implementation Details: Dynamic routing inherently introduces load-balancing challenges in distributed environments. The paper does not deeply discuss how the PI controller's state updates and dynamic thresholds impact expert-parallel communication overhead or pipeline stalls during highly distributed training.


- Baseline Comparisons: While the paper thoroughly compares DTop-p against vanilla Dense, standard Top-k, and fixed Top-p models, it lacks comparisons against other state-of-the-art dynamic routing mechanisms (such as Expert Choice Routing or methods utilizing auxiliary zero-compute experts).

---

> ### Author Rebuttal · Authors · 2026-03-28
>
> Thank you for the constructive review.
>
> > W1 & Q1 Load-balancing (LB) challenges and synchronization overhead
>
> In addition to dynamic routing, we retain a global LB loss (Section 6) to encourage balanced workloads across experts. In DTop-p, the routing dynamics are mainly at the token and layer levels, while the LB loss helps maintain balanced expert utilization under different PI-controller states, so dynamic threshold adjustment does not inherently lead to severe expert imbalance. From the implementation perspective, DTop-p is similar to token-wise Top-k expert parallelism, and can be supported by existing optimized libraries such as DeepEP and torch.grouped_mm.
>
> The additional synchronization overhead is minimal. The PI controller only requires the average number of activated experts per token after each layer’s forward pass, and this statistic is computed from the same (token, expert) assignment tensor already used for the LB loss. Therefore, the required synchronization can be performed together with the LB loss computation. The controller state and threshold update are then completed during the optimizer step using the update expression based on $e_t$​ (the velocity form of Equation 6). Overall, DTop-p introduces only minimal extra synchronization cost beyond standard MoE training.
>
> > W2 Baselines involving Expert Choice Routing or auxiliary zero-compute experts
>
> We did not include Expert Choice Routing because it is less directly applicable to autoregressive LLM training, as it relies on global token routing information. We therefore focus on token-choice MoE baselines.
>
> Methods based on auxiliary zero-compute experts are largely orthogonal to DTop-p and can be complementary in practice. Our method is designed to improve the controllability and effectiveness of Top-p routing itself, while also providing more direct compute-budget control through PI.
>
> More broadly, our baseline design was chosen to isolate the core question of the paper: whether dynamic PI control and DRN can make Top-p routing both compute-controllable and more effective than standard Top-k. Accordingly, we focus on Dense, Top-k, and fixed-threshold Top-p, together with component ablations such as DTop-p without PI, DTop-p without DRN, and Top-k + DRN.
>
> > Q2 The integral term under sudden distribution shifts / possible windup
>
> In our PI controller, the integral term accumulates past sparsity errors to reduce steady-state bias and help the routing converge to the target expert budget. As shown in Figures 2 and 3, DTop-p rapidly aligns the number of activated experts with the target and maintains a stable fluctuation range even at the beginning of training. This is further supported by the PI ablations in Figures 10 and 11: moderate PI coefficients (e.g., 0.1) adapt the threshold quickly and stably even under very different initializations, while overly large coefficients (e.g., 1) can cause larger fluctuations, and overly small coefficients adapt too slowly. In addition, the sparsity error is normalized by the total number of experts, which keeps its magnitude small and further improves controller stability.
>
> Under sudden distribution shifts, the controller may produce temporary fluctuations in the threshold and activated experts, but in our experiments, these fluctuations remain mild and do not show large spikes. In addition, the threshold is explicitly constrained to (0,1), which keeps the control signal bounded. In practical training, the batch size is also very large (e.g., over 1M tokens), making drastic token distribution shifts at the full-batch level relatively rare.
>
> > L1 Frontier scale models over 100B trained on trillions of tokens
>
> We agree that this is a limitation, as noted in Appendix H, since we do not have access to such compute resources. For reference, training a 100B+ activated-parameter MoE on 2T tokens may require over 6 months on 1024 A100 GPUs (assuming HFU = 0.2).
> That said, our current scaling study already covers models up to 2.4B active / 13.6B total parameters and up to 300B training tokens, where DTop-p consistently remains effective. We view this as a strong first validation of the robustness of the method across scaling regimes.
>
> > L2 The PI controller may require recalibration when adapted to new architectures or modalities outside standard NLP/CV transformers
>
> In our NLP and CV pre-training experiments with transformer-based architectures, DTop-p remains robust across different modalities, model sizes, and dataset sizes (Sections 5.2 and 5.3). Notably, we use the same PI coefficients across all settings, which already suggests strong robustness. This is further supported by the PI-controller ablations, which show robustness to different probability initializations and rapid threshold adaptation. Based on these results, we expect recalibration for new modalities or architectures to be limited. Moreover, DTop-p already shows clear advantages over fixed-threshold Top-p MoE with much less tuning effort.

---

> > ### Author Rebuttal · Reviewer_8kdc · 2026-04-04
> >
> > Thank you for the detailed rebuttal. All my concerns have been resolved, and I am keeping my positive score.

---

> > > ### Author Response · Authors · 2026-04-05
> > >
> > > We sincerely thank the reviewer for the constructive feedback during the review process and for the time taken to carefully evaluate our rebuttal. We are glad that our responses have addressed all of the raised concerns. We will incorporate the clarifications from our rebuttal into the final version of the paper. Thank you again for the positive and constructive engagement throughout the review process.

---

### Decision · Program_Chairs · 2026-04-30

**Decision:**

Accept (regular)

**Comment:**

This paper proposes DTop-p MoE, a dynamic Top-p routing framework for sparse MoE pre-training that combines a PI controller to keep expert activation near a target compute budget with dynamic routing normalization to allow layer-wise adaptive expert selection. It shows that fixed-threshold Top-p is unstable and hyperparameter-sensitive, and demonstrates across LLM and DiT pre-training that DTop-p improves over Top-k and fixed Top-p while matching Top-k’s average FLOPs. Overall, the authors assess a broad aspect of sparse routing in foundation models, and Overall, the study outlines a broad topic around how to make dynamic MoE routing both adaptive and compute-controllable at pre-training scale.

I recommend acceptance. The paper addresses a practical and important problem, and the proposed solution is simple, well motivated, and supported by strong empirical evidence. The main strengths are the clear diagnosis of why naive fixed Top-p is insufficient, the principled PI-based control mechanism for enforcing sparsity budgets, and the consistently positive results across NLP and vision settings, including scaling studies over expert granularity, expert capacity, model size, and dataset size.

The main reviewer concerns were about deployment details, novelty framing, and frontier-scale validation rather than any fatal weakness in the method, and the rebuttal addressed these concerns well. In particular, the authors clarified the minimal synchronization overhead, explained why PI control and DRN help beyond fixed Top-p, and provided additional evidence that the gains remain meaningful under matched FLOPs and across larger-scale settings already tested. Reviewers explicitly indicated that their concerns were resolved and kept positive scores.

Overall, this is a solid ICML paper with clear practical relevance, a sensible technical contribution, and comprehensive validation. I recommend acceptance.